# Age Variation in Patients with Troponin Level Elevation Without Obstructive Culprit Lesion or Suspected Myocardial Infarction with Non-Obstructive Coronary Arteries—Long-Term Data Covering over Decade

**DOI:** 10.3390/jcm13247685

**Published:** 2024-12-17

**Authors:** Mohammad Abumayyaleh, Clara Schlettert, Daniel Materzok, Andreas Mügge, Nazha Hamdani, Ibrahim Akin, Assem Aweimer, Ibrahim El-Battrawy

**Affiliations:** 1Department of Cardiology, Angiology, Haemostaseology and Medical Intensive Care, University Medical Center Mannheim, Medical Faculty Mannheim, Heidelberg University, Theodor-Kutzer-Ufer 1-3, 68167 Mannheim, Germany; ibrahim.akin@umm.de; 2European Center for AngioScience (ECAS) and DZHK (German Center for Cardiovascular Research (DZHK)), Partner Site, 68167 Mannheim, Germany; 3Department of Cardiology and Angiology, Bergmannsheil University Hospitals, Ruhr-University Bochum, 44789 Bochum, Germany; cschlettert@gmx.de (C.S.); d.materzok@gmx.de (D.M.); andreas.muegge@bergmannsheil.de (A.M.); assem.aweimer@bergmannsheil.de (A.A.); 4Institut für Forschung und Lehre (IFL), Department of Molecular and Experimental Cardiology, Ruhr-University Bochum, 44791 Bochum, Germany; ibrahim.elbattrawy2006@gmail.com; 5Institute of Physiology, Department of Cellular and Translational Physiology, Ruhr-University Bochum, 44801 Bochum, Germany; nazha.hamdani@rub.de

**Keywords:** myocardial infarction, MINOCA, arrhythmias, atrial fibrillation, age, cardiovascular mortality

## Abstract

**Background/Objectives:** Troponin level elevation without an obstructive culprit lesion is caused by heterogenous entities. The effect of aging on this condition has been poorly investigated. **Methods:** After screening 24,775 patients between 2010 and 2021, this study included a total of 373 patients with elevated troponin levels without an obstructive culprit lesion or suspected myocardial infarction with non-obstructive coronary arteries (MINOCAs) categorized into four age groups containing 78 patients (<51 years), 72 patients (51–60 years), 81 patients (61–70 years), and 142 patients (>70 years). This study analyzed the baseline characteristics, the in-hospital complications, in-hospital mortality, and the long-term outcomes. **Results:** The older patients exhibited a higher rate of major adverse cardiovascular in-hospital events than those of the other age groups (15.4% in the <51-year-old group vs. 36.1% in the 51–60-year-old group vs. 33.3% in the 61–70-year-old group vs. 47.2% in the >70-year-old group; *p* < 0.001). However, the rate of non-sustained ventricular tachycardia (nsVT) was higher in the 51–60-year-old patients than those of the other age groups (5.6% in the 51–60-year-old group vs. 1.3% in the 61–70-year-old group vs. 0.7% in the >70-year-old group; *p* = 0.027). At the 11-year follow-up, cardiovascular mortality was higher among the older patients compared to that of the younger patients (3.9% in the 61–70-year-old group vs. 4.2% in the >70-year-old group, *p* = 0.042), while non-cardiovascular mortality was comparable between the age groups. **Conclusions:** The older patients with troponin level elevation without an obstructive culprit lesion experienced a higher incidence of major adverse cardiovascular events during hospitalization compared to that of the younger groups. Additionally, higher cardiovascular mortality rates were revealed in the older patients at a long-term follow-up.

## 1. Introduction

Troponin level elevation without an obstructive culprit lesion is a common occurrence. This finding observed in 5–15% of patients who present with myocardial infarction (MI) [1,2]. The most frequent underlying causes of troponin level elevation without an obstructive culprit lesion appear to be coronary vasospasm, microvascular disease, coronary plaque dissection, coronary dissection, coronary thromboembolism, takotsubo cardiomyopathy, and myocarditis [2].

Whereas the treatment of MI with obstructive coronary disease by primary percutaneous coronary intervention (PCI) and novel thienopyridines has been well established, there are few randomized data on the effectiveness of preventive therapies in patients with troponin level elevation without an obstructive culprit lesion [3]. For this reason, identifying the underlying individual mechanisms to design patient-specific treatments is crucial in treating patients with troponin level elevation without an obstructive culprit lesion, ultimately resulting in optimized clinical outcomes. Choo et al. reported that treatment with renin–angiotensin system blockers and statins reduced the mortality in patients with troponin level elevation without an obstructive culprit lesion [4]. The SWEDEHEART trial showed lower rates of all-cause mortality, hospitalization for MI, ischemic stroke, and heart failure in patients with troponin level elevation without an obstructive culprit lesion treated with statins, an angiotensin-converting enzyme inhibitor (ACEI)/angiotensin receptor blocker (ARB), and a ß-blocker, but not in the patients treated with dual-antiplatelet agents [5]. However, the prognosis of this condition is not benign. Patients with troponin level elevation without obstructive coronary artery disease (CAD), as observed in the ACUITY trial, exhibited an increased risk of all cause-mortality compared to that of those with obstructive CAD at a one-year follow-up [6]. Furthermore, these patients experienced a higher incidence of heart failure (HF) comparable to that of individuals with obstructive CAD. Notably, HF in this population was predominantly characterized by preserved ejection fraction (HFpEF) [7].

Several data showed that age variation may impact the outcome of several cardiovascular diseases, e.g., HF [8]. In this regard, our aim was to provide a detailed description of demographic data, in-hospital complications, and clinical outcomes at long-term follow-up in patients with troponin level elevation without an obstructive culprit lesion grouped into different age groups.

## 2. Material and Methods

### 2.1. Study Population and Definition of Troponin Level Elevation Without Obstructive CAD

This is a monocentric, retrospective cohort study of consecutive patients admitted for troponin level elevation who underwent coronary angiography between 2010 and 2021 due to suspected acute coronary syndrome (ACS) [9]. Individuals were excluded if they met the criteria for obstructive CAD, or if their condition had a non-cardiac origin, such as stroke, pulmonary embolism, sepsis, acute respiratory distress syndrome, or end-stage renal failure [10]. Of the 24,775 patients screened, 373 patients with troponin level elevation without an obstructive culprit lesion or suspected myocardial infarction with non-obstructive coronary arteries (MINOCAs) were included in analysis. MINOCAs were defined in accordance with the 2023 ACS Guidelines of the European Society of Cardiology (ESC).

To meet the inclusion criteria, the patients first had to fulfill the modified criteria for acute MI according to the “Fourth Universal Definition of Myocardial Infarction” [11]. Additionally, there had to be clinical evidence of MI, as indicated by at least one of the following: symptoms of MI, new ischemic changes on an electrocardiogram, pathological Q waves, evidence of the new loss of a viable myocardium, new regional wall motion abnormalities suggestive of an ischemic cause, or evidence of coronary thrombus identified through angiography or autopsy. These patients were divided into four groups, younger (<51 years), early middle age (51–60 years), late middle age (61–70 years), and older (>70 years) (Figure 1). The data were collected using a specific database, including baseline characteristics, electrocardiographic data, malignant and supraventricular arrhythmias, medication given at admission and discharge, echocardiographic data at discharge, in-hospital complications, and in-hospital mortality. The clinical outcomes, including stroke, thromboembolic events, the recurrence of elevation troponin levels, cardiac arrest, percutaneous coronary intervention (PCI), and mortality due to cardiac or non-cardiac causes at a long-term follow-up, were gathered. This study was executed in compliance with the Declaration of Helsinki regarding investigations on human subjects, and this study protocol was approved by the Ethics Committee of the participating institutions.

### 2.2. Definition of Troponin Level Elevation Without an Obstructive Culprit Lesion

Coronary angiography was performed by interventional cardiologists during admission for all the screened patients with elevated troponin levels and suspected ACS following the 2023 ACS guidelines. Obstructive CAD was defined as the presence of one or more coronary stenoses with a diameter reduction of ≥50%. Non-obstructive CAD was further classified based on the angiographic findings as stenoses <50%, or no angiographic evidence of coronary disease. Revascularization strategies were implemented in cases where a culprit lesion was identified or when the visually estimated diameter stenosis severity was ≥70% in non-left main coronary artery disease or ≥50% in left main coronary artery disease. These strategies were pursued when coronary artery bypass grafting (CABG) therapy was not preferred.

### 2.3. High-Sensitivity Troponin I

Regarding troponin analysis, high-sensitivity cardiac troponin I (hs-TnI) assays were used. The 99th percentile values were identified for all the patients using the results of measurements from a healthy reference population [12]. The first hs-Tn test was conducted upon admission, with the second test scheduled within the initial three hours, as recommended by the ESC [13,14].

### 2.4. Outcomes

We evaluated cardiovascular and non-cardiovascular mortality as the primary endpoints during the long-term follow-up. The in-hospital complications evaluated in this study included HF, cardiac arrest, left ventricular thrombus, pulmonary edema, cardiogenic shock, extracorporeal membrane oxygenation (ECMO), invasive ventilation, non-invasive ventilation, stroke, malignant cardiac arrhythmias, and supraventricular arrhythmias. The clinical outcomes at the long-term follow-up, including stroke, thromboembolic event, the recurrence of troponin level elevation, cardiac arrest, and PCI as the secondary endpoints, were also assessed. The mean duration of the long-term follow-up was 6.2 ± 3.1 years. Major adverse cardiovascular events were defined as a composite of cardiovascular death, myocardial infarction, or ischemic stroke.

### 2.5. Statical Analysis

Descriptive und comparative analyses were performed. Categorial variables are presented as numbers (frequency) and percentages. Continuous variables are presented as mean ± standard deviation (SD) if the distribution was normal, or the median (interquartile range) if not. Comparisons of the categorial variables were conducted with the Pearson chi-square test. For continuous variables, the Kruskal–Wallis test with Dunn’s multiple-comparison test were used. The comparative analysis of quantitative variables was presented using the Mann–Whitney U test for non-parametric variables and the *t*-student test for parametric variables, as verified by the Shapiro–Wilk test. All the tests were 2-sided, and a *p* value < 0.05 was considered statistically significant. SPSS version 26.0 (IBM Corp., Armonk, NY, USA) was used for statistical analyses.

## 3. Results

### 3.1. Baseline Characteristics According to Age

A total of 373 patients with troponin level elevation without an obstructive culprit lesion were included in this study, with a mean age of 63 ± 15.6 years old, and 49.6% were male. Out of the total sample, 78 (20.9%) were <51 years, 72 (19.3%) were aged from 51 to 60 years, 81 (21.7%) were aged from 61 to 70 years, and 142 (38%) were > 70 years old. Males were more prevalent among the patients < 51 years old (age < 51 years: 70.5%, 51–60 years: 58.3%, 61–70 years: 44.4%, and >70 years: 36.6%; *p* < 0.001). The baseline characteristics are presented in Table 1.

According to their medical history, the older patients had higher rates of arterial hypertension (42.3% in the <51-year-old group vs. 51.4% in 51–60-year-old group vs. 81.5% in the 61–70-year-olds vs. 83.1% in the >70 group; *p* < 0.001), malignancy (2.6% vs. 2.9% vs. 15% vs. 21.8%; *p* < 0.001), and kidney disease (6.4% vs. 7% vs. 14.8% vs. 21.8%; *p* = 0.003). The number of current smokers was higher in the <51-year-old group than those in the other groups (age < 51: 40.3% vs. age 51–60: 25.4% vs. age 61–70: 30.9% vs. age > 70: 7.9%; *p* < 0.001). The patients in the 51–60- and 61–70-year-old groups displayed psychiatric disorders more frequently than the other groups (7.7% vs. 14.1% vs. 17.3% vs. 6.3%; *p* = 0.045). The prevalence of atrial fibrillation (AF) on admission was higher in the >70-year-old group as compared to that of the other groups (1.3% vs. 8.6% vs. 16.1% vs. 26.1%; *p* < 0.001). The left ventricular ejection fraction (LVEF) was higher in the <51-year-old group as compared to those of the other groups (46.7% ± 23 vs. 37.5% ± 25 vs. 34.4% ± 27 vs. 32.6% ± 25; *p* < 0.001). The arrhythmias, the symptoms, the laboratory values, and the medication given on admission are presented in Table 1. The medication given at discharge is presented in Table 2.

### 3.2. In-Hospital Complications Presented According to Age

Major adverse cardiovascular events were associated with increased age (15.4% vs. 36.1% vs. 33.3% vs. 47.2%; *p* < 0.001). Interestingly, non-sustained ventricular arrhythmias (nsVTs) occurred more frequently in the 51–60-year-old group as compared to those in the other age groups (0% vs. 5.6% vs. 1.3% vs. 0.7; *p* = 0.027). Newly diagnosed AF was more prevalent in the >70 age group as compared to the rates in the other groups (3.8% vs. 6.9% vs. 7.4% vs. 20.4%; *p* < 0.001). The other cardiovascular in-hospital complications, including mortality, were similar in all the age groups (in-hospital mortality: none vs. 4.2% vs. 2.5% vs. 3.5%; *p* = 0.371) (Figure 2). Table 3 presents the in-hospital-complications according to age.

### 3.3. LVEF and Clinical Outcomes Presented According to Age at Long-Term Follow-Up

The LVEF at the follow-up was higher in the <51 age group than those in the other groups (58.1% ± 11 vs. 48.4% ± 17 vs. 41.3% ± 14 vs. 43.1% ± 12; *p* < 0.001) (Table 4). The rate of major adverse cardiovascular events at the long-term follow-up was higher in the >70 age group, without statistical significance (6.5% vs. 21.6% vs. 26.2% vs. 61.1%; *p* = 0.126). Cardiovascular mortality was more common in the patients >70 years of age (none in the <51- and 51–60-year-olds vs. 3.9% in the 61–70-year-olds vs. 4.2% in the >70-year-olds; *p* = 0.042). Non-cardiovascular mortality was more frequent in the patients aged 51–60 and >70 years, but without statistically significance (none in the <51-year-olds vs. 8.2% in the 51–60-year-olds vs. 3.9% in the 61–70-year-olds vs. 8.5% in the >70-year-olds; *p* = 0.075) (Figure 3). The clinical outcomes are presented in Table 5.

FU EF, follow up ejection fraction; LVEF, left ventricular ejection fraction; Bold indicates significant values.

## 4. Discussion

The main findings of this study are as follows: (1) The incidence of troponin level elevation without an obstructive culprit lesion was higher in the patients aged >70 years compared to those of the other age groups. (2) Adverse in-hospital events were more frequent in the older patients, but the in-hospital mortality rates were similar across the age groups. (3) At the long-term follow-up, cardiovascular mortality was higher in the group aged >70 years. (4) The overall all-cause mortality rate among all the patients over a decade was 24.1%.

Age impacts the outcome of cardiovascular diseases, such as atherosclerosis and arterial hypertension, and is a well-established risk factor for worse prognoses in these conditions [15]. Troponin level elevation without an obstructive culprit lesion presents an inherent challenge due to the numerous possible etiologies and pathogenic mechanisms. In this study, troponin level elevation without an obstructive culprit lesion was characterized by clinical evidence of MI with normal coronary arteries or stenosis <50%. However, coronary microvascular disease (CMD) plays a significant role in this condition [1]. In this context, the functional assessment of microcirculation represents a promising technique for evaluating patients with troponin level elevation, given the high prevalence of CMD and its significant clinical implications. This approach is likely to become more widely used in the future.

In our study, the incidence of troponin level elevation without an obstructive culprit lesion was notably high in the patients >70 years old. Among the patients <51 years old, males were more frequently affected; half of the total cohort consisted of females. One study corroborated our findings, showing a high prevalence of troponin level elevation without an obstructive culprit lesion in older patients, with female being the predominant sex [16]. Conversely, a systemic review reported that younger, predominantly female patients were more likely to experience troponin level elevation without an obstructive culprit lesion [17]. Another study highlighted that woman aged ≤70 years had worse prognoses, with significantly higher rates of major adverse cardiovascular events (MACEs) compared to men and women with MI and obstructive CAD [18]. This was attributed to distinct pathophysiological mechanisms affecting the long-term outcomes [18]. Additionally, the data from the VIRGO registry indicated that females had a fivefold higher risk of this developing syndrome compared to that of males [19].

Our data suggest that troponin level elevation without an obstructive culprit lesion becomes more prevalent with advancing age. Thus, a conservative approach avoiding invasive diagnostic measures may be appropriate in suspected cases of this syndrome. Furthermore, we observed a higher prevalence of psychiatric disorders in the 51–60 and 61–70 age groups. This is consistent with previous findings indicating that patients with troponin level elevation without an obstructive culprit lesion have a higher rate of psychiatric illness compared to those with obstructive CAD [20]. Psychiatric disorders have also been more frequently associated with takotsubo syndrome (TTS) [21]. Moreover, anxiety has been observed in patients with elevated troponin levels, which is linked to an increased risk of all-cause mortality [22]. In the 51–60 age group, the incidence of non-sustained ventricular tachycardia (nsVT) was higher compared to that in the other groups. This may be attributed to the higher prevalence of TTS and myocarditis in this age group [23,24]. Additionally, the patients in this group exhibited a reduced LVEF and elevated troponin levels, both of which are associated with an increased risk of ventricular arrhythmias. However, the data on troponin level elevation without an obstructive culprit lesion across different age groups remain limited.

Adverse events, including AF, were more frequent in the individuals >70 years. A meta-analysis reported that patients with troponin level elevation without an obstructive culprit lesion are at a high risk of having an MACE [25]. This increased risk may be due to undertreatment and the lack of personalized therapeutic options for these patients. Additionally, they often present with left ventricular dysfunction [26]. AF has also been identified as a risk factor for MACEs in patients with elevated troponin levels without an obstructive culprit lesion [27]. In our study, in-hospital mortality was comparable across all the age groups. However, unlike MI caused by obstructive CAD in older patients, troponin level elevation without obstructive CAD is associated with a higher risk of in-hospital mortality, as reported by Ishii et al. [28].

Our study revealed that the patients >70 years old exhibited a higher cardiovascular mortality rate during the long-term follow-up compared to those of the other age groups. Over a decade, the overall all-cause mortality rate among all the patients was 24.1%. This rate is higher compared to the previously reported data, such as in the VIRGO registry, which documented a 12-month mortality rate of 0.6% for patients with troponin level elevation without obstructive CAD. Similarly, a multicenter cohort study reported a 20% mortality rate at a 5-year follow-up, while a systematic meta-analysis estimated a mortality rate of 2% at 25 months [29,30]. Another meta-analysis reported a 4.7% mortality rate at 12 months [17]. In another study, approximately half of the patients with a history of troponin level elevation without an obstructive culprit lesion who experienced reinfarction showed evidence of atherosclerosis progression, often requiring revascularization. Furthermore, the recommended low-density-lipoprotein cholesterol (LDL-C) target levels were achieved in only a minority of these patients, suggesting that the cardiovascular disease risk in this population may be underestimated or insufficiently managed. It is important to note that our dataset spans a decade and provides a broader context for evaluating mortality trends. However, the patients >70 years old in our study exhibited a higher prevalence of cardiovascular comorbidities, including AF, and presented with a lower LVEF at admission compared to those of the other age groups. This indicates a more severe cardiovascular risk profile in the elderly population. Currently, data on the age-related trends in patients with elevated troponin levels without an obstructive culprit lesion extending beyond ten years remain scarce.

Summarizing, the prevalence of troponin level elevation without an obstructive culprit lesion was observed to be higher in the patients >70 years old compared to that in the other age groups. This group experienced more adverse events and had a higher cardiovascular mortality rate than those of the other patients. In this context, implementing advanced diagnostic approaches, such as cardiac magnetic resonance (CMR) imaging and the functional assessment of microcirculation, may be beneficial in selected cases, particularly at initial presentation with suspected non-obstructive coronary disease. However, the feasibility and clinical utility of such approaches require further investigation. The more refined stratification of patients with troponin level elevation without an obstructive culprit lesion may also help in tailoring treatment strategies. While age remains an important consideration in this population, additional studies are essential to better characterize and address their specific clinical needs.

## 5. Limitations

Several limitations of this study should be noted. This is a monocentric retrospective study. CMR imaging was not performed systematically on all the patients. Angiographic classification was assessed using visual evaluation. Intravascular imaging, which represents an important diagnostic tool, was not performed, including intravascular ultrasound (IVUS) or optical coherence tomography (OCT).

## Figures and Tables

**Figure 1 jcm-13-07685-f001:**
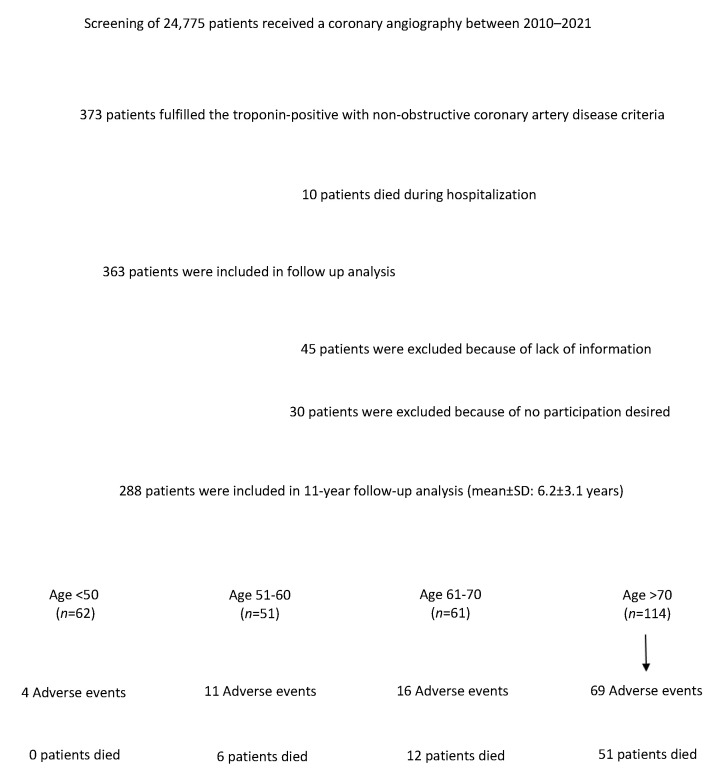
A flow-chart presenting the screened data and the patients included in the present study.

**Figure 2 jcm-13-07685-f002:**
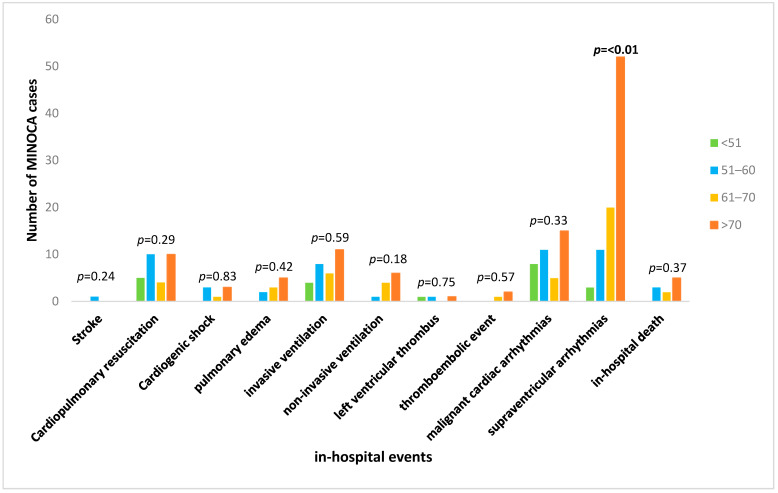
In-hospital events related to age.

**Figure 3 jcm-13-07685-f003:**
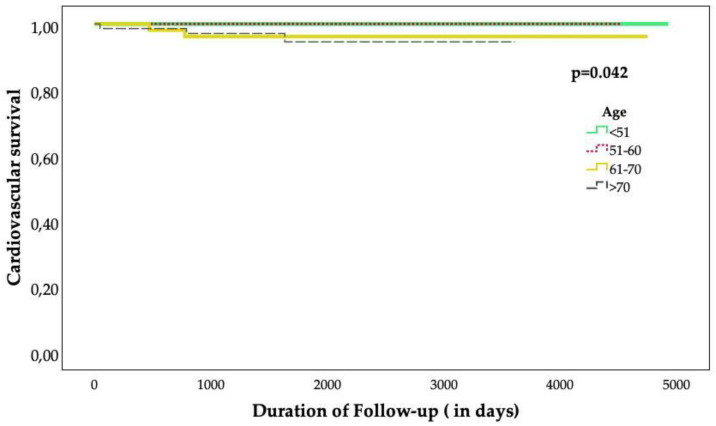
Kaplan–Meier curve.

**Table 1 jcm-13-07685-t001:** Baseline characteristics of 373 patients initially presenting elevated troponin levels with non-obstructive coronary artery disease presented according to age.

Variables	All Patients *n* = 373	<51 *n* = 78	51–60 *n* = 72	61–70 *n* = 81	>70 *n* = 142	*p* Value
Age—years, mean ± SD	63 ± 15.6	40 ± 9.3	56 ± 3	65 ± 3	78 ± 5	**<0.001**
Male—*n* (%)	185 (49.6)	55 (70.5)	42 (58.3)	36 (44.4)	52 (36.6)	**<0.001**
BMI—kg/m^2^, mean ± SD	27.6 ± 5.6	27.8 ± 6	28.4 ± 6	27.3 ± 5	27.3 ± 5	0.502
Duration of hospitalization—days, mean ± SD	10 ± 8.5	8 ± 5	10.2 ± 8	11 ± 12	10.7 ± 8	0.060
Symptoms—*n* (%)						
Angina pectoris	226 (61.6)	55 (71.4)	41 (58.6)	49 (60.5)	81 (58.3)	0.252
Dyspnea	164 (44.4)	18 (23.4)	34 (47.2)	39 (48.2)	73 (52.5)	**<0.001**
Palpations	45 (12.3)	12 (15.6)	7 (10)	9 (11.1)	17 (12.2)	0.750
Clinic parameter, mean ± SD						
Systolic BP, mmHg	146.6 ± 64.4	137.1 ± 26	146.2 ± 31	154.3 ± 126	147 ± 26	0.553
Diastolic BP, mmHg	84.8 ± 18.1	58.7 ± 17	90.9 ± 24	81.4 ± 14	83.2 ± 17	**0.026**
Heart rate, bpm	89.8 ± 28.6	88.3 ± 24	93.8 ± 27	88.1 ± 33	89.6 ± 29	0.609
ECG—*n* (%)						
ST elevation	55 (14.8)	25 (32.1)	7 (9.7)	7 (8.6)	16 (11.3)	**<0.001**
Inversed T-Waves	183 (49.2)	27 (34.6)	33 (45.8)	44 (54.3)	79 (56)	**0.015**
Medical history—*n* (%)						
Current Smoking	85 (23.1)	31 (40.3)	18 (25.4)	25 (30.9)	11 (7.9)	**<0.001**
Obesity	112 (30)	23 (29.5)	24 (34.3)	25 (30.9)	40 (28.2)	0.887
Arterial hypertension	253 (68.2)	33 (42.3)	36 (51.4)	66 (81.5)	118 (83.1)	**<0.001**
Dyslipidemia	99 (26.6)	16 (20.5)	19 (26.8)	22 (27.2)	42 (29.6)	0.547
Diabetes Mellitus	65 (17.5)	8 (10.3)	10 (14.1)	13 (16.1)	34 (23.9)	0.054
COPD	47 (12.6)	0 (0)	8 (11.3)	18 (22.2)	21 (14.8)	**<0.001**
Bronchial Asthma	33 (8.9)	7 (9)	10 (14.1)	5 (6.2)	11 (7.7)	0.344
Malignancy	47 (12.7)	2 (2.6)	2 (2.9)	12 (15)	31 (21.8)	**<0.001**
Kidney disease	53 (14.3)	5 (6.4)	5 (7)	12 (14.8)	31 (21.8)	**0.003**
Neurological disease	90 (24.3)	11 (14.1)	12 (17.1)	19 (23.5)	48 (33.8)	**0.004**
Autoimmune disease	17 (4.6)	5 (6.4)	2 (2.8)	3 (3.7)	7 (5)	0.733
Psychiatric disease	39 (10.5)	6 (7.7)	10 (14.1)	14 (17.3)	9 (6.3)	**0.041**
Pacemaker	14 (3.8)	0 (0)	1 (1.4)	2 (2.5)	11 (7.8)	**0.012**
History of Malignant cardiac arrhythmias *—*n* (%)	8 (2.2)	0 (0)	0 (0)	2 (2.5)	6 (7.8)	0.101
Bradycardiac arrhythmias	6 (1.6)	0 (0)	0 (0)	1 (1.2)	5 (3.6)	0.123
-AV block 2 Mobitz	3 (0.8)	0 (0)	0 (0)	1 (1.2)	2 (1.4)	0.567
-AV block 3	3 (0.8)	0 (0)	0 (0)	0 (0)	3 (2.1)	0.179
-Asystole	0 (0)	0 (0)	0 (0)	0 (0)	0 (0)	0
Ventricular arrhythmias	1 (0.3)	0 (0)	0 (0)	1 (1.2)	0 (0)	0.311
-sustained	1 (0.3)	0 (0)	0 (0)	1 (1.2)	0 (0)	0.311
-non-sustained	0 (0)	0 (0)	0 (0)	0 (0)	0 (0)	0
-ventricular fibrillation	0 (0)	0 (0)	0 (0)	0 (0)	0 (0)	0
Torsades de pointes	1 (0.3)	0 (0)	0 (0)	0 (0)	1 (0.7)	0.658
Atrial fibrillation *	57 (15.4)	1 (1.3)	6 (8.6)	13 (16.1)	37 (26.1)	**<0.001**
Laboratory values, mean ± SD						
Troponin (µg/L)	2.1 ± 8.4	2.8 ± 5.8	2.7 ± 12.1	1.3 ± 7.3	1 ± 5	0.234
Creatin Phosphokinase (U/L)	288.6 ± 548.9	352.2 ± 407	427.4 ± 943	235.4 ± 340	215 ± 418	**0.030**
BNP (pg/ml)	433.5 ± 1004.5	175 ± 390	432 ± 1286	560 ± 987	514 ± 1070	0.181
Creatinine (mg/dl)	3.3 ± 30.9	0.9 ± 0.2	12.9 ± 70.6	1.1 ± 0.4	1.1 ± 0.4	**0.040**
TSH (mU/l)	2 ± 1.7	2 ± 1.2	2.3 ± 2.4	1.8 ± 1.8	1.9 ± 1.4	0.986
fT3 (pmol/l)	5.2 ± 1	5.5 ± 1	5.2 ± 1	5.2 ± 1.1	4.8 ± 0.9	0.861
fT4 (ng/l)	10.6 ± 2.8	10.2 ± 1.8	9.1 ± 1.5	10.6 ± 3.1	12 ± 3.3	0.941
Echocardiography data, *n* (%)						
LVEF %, (on admission), mean ± SD	36.9 ± 25.7	46.7 ± 23	37.5 ± 25	34.4 ± 27	32.6 ± 25	**<0.001**
-LVEF > 50 %	170 (66.9)	52 (82.5)	29 (60.4)	35 (67.3)	54 (59.3)	**0.016**
-LVEF 40–49 %	31 (12.2)	3 (4.8)	9 (18.8)	5 (9.6)	14 (15.4)	0.096
-LVEF < 40 %	52 (20.5)	8 (12.7)	10 (20.8)	11 (21.2)	23 (25.3)	0.305
Left ventricular hypertrophy	103 (29.1)	15 (32.5)	20 (30.3)	23 (29.5)	45 (33.8)	0.177
Tricuspid valve regurgitation	89 (24.2)	7 (9)	15 (21.4)	21 (26.3)	46 (32.9)	**0.001**
-mild	63 (17.1)	6 (7.7)	11 (15.7)	14 (17.5)	32 (22.9)	0.041
-moderate	23 (6.3)	1 (1.3)	4 (5.7)	6 (7.5)	12 (8.6)	0.187
-severe	3 (0.8)	0 (0)	0 (0)	1 (1.3)	2 (1.4)	0.564
Mitral valve regurgitation	106 (28.8)	7 (9)	16 (22.9)	21 (26.3)	62 (44.3)	**<0.001**
-mild	77 (20.9)	6 (7.7)	12 (17.1)	17 (21.3)	42 (29.8)	**0.001**
-moderate	19 (5.2)	1 (1.3)	2 (2.9)	3 (3.8)	13 (9.2)	**0.041**
-severe	10 (2.7)	0 (0)	2 (2.9)	1 (1.3)	7 (5)	0.137
Aortic valve regurgitation	39 (10.6)	0 (0)	7 (10)	3 (3.8)	29 (20.7)	**<0.001**
-mild	31 (8.4)	0 (0)	5 (7.1)	3 (3.8)	23 (16.4)	**<0.001**
-moderate	4 (1.1)	0 (0)	2 (2.8)	0 (0)	2 (1.4)	0.271
-severe	4 (1.1)	0 (0)	0 (0)	0 (0)	4 (2.9)	0.086
Drugs on admission, *n* (%)						
ß-Blocker	131 (35.2)	8 (10.3)	20 (28.2)	32 (39.5)	71 (50.7)	**<0.001**
ACE inhibitor	121 (32.6)	14 (17.9)	17 (23.9)	35 (43.2)	55 (39)	**<0.001**
Sartans	57 (15.3)	4 (5.1)	9 (12.7)	11 (13.6)	33 (23.6)	**0.003**
Ca-Blocker	74 (19.9)	6 (7.7)	8 (11.3)	16 (19.8)	44 (31.4)	**<0.001**
Diuretics	101 (27.2)	8 (10.3)	14 (19.7)	16 (19.8)	63 (45)	**<0.001**
Statin	281 (76.8)	78 (27.1)	75 (25.2)	61 (21.3)	41 (18.1)	**<0.001**
Anticoagulants **	58 (15.6)	1 (1.3)	2 (2.8)	17 (21)	38 (27.1)	**<0.001**
Aspirin	79 (21.2)	7 (9)	12 (16.9)	23 (28.4)	37 (26.4)	**0.006**
Clopidogrel	18 (4.8)	2 (2.6)	2 (2.8)	6 (7.4)	8 (5.7)	0.416
Prasugrel	0 (0)	0 (0)	0 (0)	0 (0)	0 (0)	0
Antiarrhythmics ***	10 (2.7)	0 (0)	3 (4.2)	3 (3.7)	4 (2.9)	0.375

*p* values for the comparison between groups of ages; SD, Standard deviation; BMI, body mass index; BP, blood pressure; ECG, Electrocardiogram; COPD, Chronic obstructive pulmonary disease; AV, atrioventricular; BNP, brain natriuretic Peptide; LV EF, Ejection fraction; ACE, Angiotensin-converting-enzyme; *, only one malignant cardiac/supraventricular arrhythmia is counted per patient (even if one patient has several arrhythmias at the same time); ** coumarin, heparin, selective factor 10-blocker, direct thrombin inhibitors; ***, Ivabradine, Flecainide, Sotalol, Dronedarone, Digitalis; Bold indicates significant value.

**Table 2 jcm-13-07685-t002:** Medication given at discharge.

Variables	All Patients *n* = 373	<51 *n* = 78	51–60 *n* = 72	61–70 *n* = 81	>70 *n* = 142	*p* Value
ß-Blocker, *n* (%)	270 (72.4)	50 (64.1)	54 (75)	63 (77.8)	103 (72.5)	0.252
ACE inhibitor, *n* (%)	224 (60.1)	46 (59)	47 (65.3)	51 (63)	80 (56.3)	0.584
Sartans, *n* (%)	62 (16.6)	6 (7.7)	10 (13.9)	15 (18.5)	31 (21.8)	**0.049**
Ca Blocker, *n* (%)	100 (26.8)	12 (15.4)	15 (20.8)	24 (29.6)	49 (34.5)	**0.011**
Diuretics, *n* (%)	166 (44.5)	13 (16.7)	29 (40.3)	34 (42)	90 (63.4)	**<0.001**
Statin, *n* (%)	206 (55.2)	21 (26.9)	32 (44.4)	41 (50.6)	73 (51.4)	**<0.001**
Anticoagulants *, *n* (%)	104 (27.9)	3 (3.9)	17 (23.6)	25 (30.9)	59 (41.5)	**<0.001**
Aspirin, *n* (%)	175 (46.9)	27 (34.6)	32 (44.4)	46 (56.8)	70 (49.3)	**0.038**
Clopidogrel, *n* (%)	68 (18.2)	14 (18)	11 (15.3)	20 (24.7)	23 (16.2)	0.381
Prasugrel, *n* (%)	0 (0)	0 (0)	0 (0)	0 (0)	0 (0)	0
Antiarrhythmics **, *n* (%)	26 (7)	1 (1.3)	7 (9.7)	5 (6.2)	13 (9.2)	0.119

ACE, Angiotensin-converting-enzyme; * Coumarin, Heparin, selective factor 10-blocker, direct thrombin inhibitors; ** Ivabradine, Flecainide, Sotalol, Dronedarone, Digitalis; Bold indicates significant value.

**Table 3 jcm-13-07685-t003:** In-hospital complications presented according to age.

Variables	All Patients *n* = 373	<51 *n* = 78	51–60 *n* = 72	61–70 *n* = 81	>70 *n* = 142	*p* Value
Adverse event, *n* (%)	132 (35.5)	12 (15.4)	26 (36.1)	27 (33.3)	67 (47.2)	**<0.001**
CPR	7 (1.9)	0 (0)	3 (4.2)	1 (1.2)	3 (2.1)	0.290
Left ventricular thrombus	3 (0.8)	1 (1.3)	1 (1.4)	0 (0)	1 (0.7)	0.754
Thromboembolic event	3 (0.8)	0 (0)	0 (0)	1 (1.2)	2 (1.4)	0.564
Pulmonary edema	9 (2.7)	1 (1.3)	2 (2.8)	2 (2.5)	4 (2.8)	0.906
Cardiogenic shock	9 (2.7)	1 (1.3)	2 (2.8)	2 (2.5)	4 (2.8)	0.906
Invasive ventilation	29 (7.8)	4 (5.1)	8 (11.1)	6 (7.4)	11 (7.8)	0.597
Non-invasive ventilation	11 (3)	0 (0)	1 (1.4)	4 (4.9)	6 (4.3)	0.180
Stroke	1 (0.3)	0 (0)	1 (1.4)	0 (0)	0 (0)	0.243
Malignant cardiac arrhythmias *, *n* (%) (on admission/in hospital)	39 (10.5)	8 (10.3)	11 (15.3)	5 (6.2)	15 (10.6)	0.339
Bradycardiac arrhythmias	13 (3.5)	3 (3.8)	1 (1.4)	1 (1.2)	8 (5.6)	0.248
-AV block 2 Mobitz	3 (0.8)	2 (2.6)	0 (0)	0 (0)	1 (0.7)	0.234
-AV block 3	1 (0.3)	0 (0)	0 (0)	0 (0)	1 (0.7)	0.655
-Asystole	10 (2.7)	2 (2.6)	1 (1.4)	1 (1.2)	6 (4.2)	0.494
Ventricular arrhythmias	14 (3.8)	2 (2.6)	5 (6.9)	4 (4.9)	5 (3.5)	0.438
-sustained	8 (2.2)	2 (2.6)	2 (2.8)	0 (0)	4 (2.8)	0.524
-non-sustained	6 (1.6)	0 (0)	4 (5.6)	1 (1.3)	1 (0.7)	**0.027**
-ventricular fibrillation	16 (4.3)	3 (3.8)	7 (9.7)	3 (3.7)	3 (2.1)	0.075
Torsades de pointes	2 (0.5)	0 (0)	1 (1.4)	0 (0)	1 (0.7)	0.598
Supraventricular arrhythmias *, *n* (%)	85 (22.8)	3 (3.8)	11 (15.3)	19 (23.5)	52 (36.6)	**<0.001**
Atrial fibrillation	77 (20.6)	3 (3.8)	8 (11.1)	15 (18.5)	51 (35.9)	**<0.001**
-first appearance	43 (11.5)	3 (3.8)	5 (6.9)	6 (7.4)	29 (20.4)	**<0.001**
-recurrens	34 (9.1)	0 (0)	3 (4.2)	9 (11.1)	22 (15.5)	**<0.001**
Atrial flutter	9 (2.4)	0 (0)	3 (4.2)	4 (4.9)	2 (1.4)	0.129
-first appereance	9 (2.4)	0 (0)	3 (4.2)	4 (4.9)	2 (1.4)	0.129
-recurrens	0 (0)	0 (0)	0 (0)	0 (0)	0 (0)	0
In-hospital death, *n* (%)	10 (2.7)	0 (0)	3 (4.2)	2 (2.5)	5 (3.5)	0.371
Cardiac caused death	6 (1.6)	0 (0)	2 (2.8)	1 (1.3)	3 (2.1)	0.535
Non-cardiac caused death	4 (1.1)	0 (0)	1 (1.4)	1 (1.3)	2 (1.4)	0.783

adverse event, major adverse cardiovascular events; CPR, cardiopulmonary resuscitation; AV, atrioventricular; *, only one malignant cardiac/supraventricular arrhythmia is counted per patient (even if one patient has several arrhythmias at the same time); Bold indicates significant values.

**Table 4 jcm-13-07685-t004:** EF at follow-up and during in-hospital treatment.

Variables	All Patients *n* = 373	<51 *n* = 78	51–60 *n* = 72	61–70 *n* = 82	>70 *n* = 142	*p* Value
FU EF (%), mean ± SD	49 ± 14.7	58.1 ± 11	48.4 ± 17	42.3 ± 14	43.1 ± 12	**<0.001**
-LVEF ≥ 50 %, *n* (%)	184 (49.3)	33 (42.3)	21 (30)	15 (19.5)	31 (23)	**0.005**
-LVEF 40-49 %, *n* (%)	100 (27.8)	3 (3.9)	7 (10)	5 (6.5)	12 (8.9)	0.457
-LVEF < 40 %, *n* (%)	27 (7.5)	5 (6.4)	10 (14.3)	10 (13)	25 (18.5)	0.106
Cardioversion, *n* (%)	50 (13.9)	5 (6.4)	14 (19.4)	17 (21.3)	28 (19.7)	**0.042**
Pacemaker, *n* (%)	64 (17.2)	2 (2.6)	8 (11.1)	5 (6.2)	9 (6.3)	0.208
LifeVest, *n* (%)	24 (6.5)	5 (6.4)	4 (5.6)	1 (1.2)	2 (1.4)	0.097

**Table 5 jcm-13-07685-t005:** Clinical outcomes presented according to age at follow-up.

	All Patients *n* = 288	<51 *n* = 62	51–60 *n* = 51	61–70 *n* = 61	>70 *n* = 114	*p* Value
Adverse event, *n* (%)	100 (34.8)	4 (6.5)	11 (21.6)	16 (26.2)	69 (61.1)	0.126
Stroke	10 (4.2)	2 (3.9)	1 (2.1)	1 (2)	6 (6.8)	0.797
Thromboembolic event	6 (2.5)	2 (3.2)	0 (0)	1 (1.9)	3 (4)	0.385
Recurrence of Troponin-positive with non-obstructive CAD	3 (1.3)	0 (0)	1 (2.1)	1 (1.9)	1 (1.4)	0.981
Cardiac arrest	4 (1.7)	0 (0)	1 (2.1)	1 (1.9)	2 (2.7)	0.900
Percutaneous coronary intervention	17 (7.2)	2 (3.2)	3 (6.3)	3 (5.7)	9 (12.3)	0.352
Death, *n* (%)	69 (24.1)	0 (0)	6 (11.8)	12 (20)	51 (45.1)	0.067
-cardiac caused death	5 (2.1)	0 (0)	0 (0)	2 (3.9)	3 (4.2)	**0.042**
-non-cardiac caused death	12 (5.1)	0 (0)	4 (8.2)	2 (3.9)	6 (8.5)	0.075

Adverse event, major adverse cardiovascular events; CAD, coronary artery disease; Bold indicates significant values.

## Data Availability

The datasets generated and analyzed during the current study are available from the corresponding author on reasonable request.

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
