# Peer review of "Age Variation in Patients with Troponin Level Elevation Without Obstructive Culprit Lesion or Suspected Myocardial Infarction with Non-Obstructive Coronary Arteries—Long-Term Data Covering over Decade"

_jcm, 2024, doi:10.3390/jcm13247685_

Round 1
Reviewer 1 Report
Comments and Suggestions for Authors
The authors evaluated the impact of age on patients with elevated troponin without an obstructive lesion, analyzing data from 373 individuals. They found that older patients had a higher rate of major adverse cardiovascular events during hospitalization compared to younger groups. Specifically, non-sustained ventricular tachycardia was more prevalent in patients aged 51 to 60 years. Additionally, long-term follow-up indicated increased cardiovascular mortality in older patients compared to younger ones.
Congratulations to the authors on this interesting work addressing a highly relevant topic.
Issues/Suggestions:
Methods: The inclusion criteria should be improved, particularly regarding the following points:
· The authors screened all patients with chest pain or diagnosed acute coronary syndrome. Which guidelines were followed for this process?
· Including the criteria for defining MINOCA according to the European Society of Cardiology guidelines would be beneficial.
· Was cardiac magnetic resonance imaging performed? CMR is necessary to determine the ischemic nature of myocardial damage and confirm the diagnosis of MINOCA. Additionally, CMR is crucial for prognostic purposes, as it identifies three phenotypes with differing prognoses: late gadolinium enhancement positive/mapping positive, late gadolinium enhancement negative/mapping positive, and normal CMR. Expanding the bibliography and discussion on this topic would be useful.
· How were the age cut-offs determined for the population division? I suggest also considering a division based on the median age of the population.
How do the authors explain such a low left ventricular ejection fraction upon admission? Were other categories of patients included (e.g., takotsubo)?
The population was largely under-treated. Could this have influenced the outcomes? I recommend expanding the discussion on this point, considering that patients with MINOCA are at high risk for reinfarction and that managing low-density lipoprotein levels is mandatory (cite: 10.1016/j.pcad.2023.10.006).
Finally, considering the study's limitations and design, I recommend toning down the conclusions and interpretation of the results.
Comments on the Quality of English LanguagePerform a thorough proofreading to correct grammatical issues and ensure clarity and readability.
Author Response
Issues/Suggestions:
Methods: The inclusion criteria should be improved, particularly regarding the following points:
- The authors screened all patients with chest pain or diagnosed acute coronary syndrome. Which guidelines were followed for this process?
Response:
- Any case with a positive troponin value without significant coronary stenosis was screened. Troponin was determined in cases of suspected acute coronary syndrome (ACS). A total of 373 patients fulfilled the inclusion criteria. To meet these criteria, patients first had to meet the modified criteria for acute myocardial infarction (AMI) according to the "Fourth Universal Definition of Myocardial Infarction". First, there had to be an increase or decrease in cardiac troponin levels, with at least one value above the 99th percentile in the laboratory. Secondly, there had to be clinical evidence of MI, as evidenced by at least one of the following conditions Symptoms of myocardial ischemia, new ischemic changes on the electrocardiogram, pathological Q waves, evidence of new loss of viable myocardium or new regional wall motion abnormalities suggesting an ischemic cause, or evidence of coronary thrombus by angiography or autopsy. As a secondary criterion, coronary artery obstruction had to be excluded in patients undergoing angiography. This was the case if the stenosis in any of the major epicardial vessels was <50% and no subsequent intervention was performed. ACS Guidelines 2023 have been followed as a reference for this manuscript. We revised and improved the material and methods section as required.
- Including the criteria for defining MINOCA according to the European Society of Cardiology guidelines would be beneficial.
Response:
- Thank you for your suggestion. The criteria for defining myocardial infarction with non-obstructive coronary arteries (MINOCA) are outlined in the Methods section of the manuscript, where we referenced the 2023 ACS guidelines.
- Was cardiac magnetic resonance imaging performed? CMR is necessary to determine the ischemic nature of myocardial damage and confirm the diagnosis of MINOCA. Additionally, CMR is crucial for prognostic purposes, as it identifies three phenotypes with differing prognoses: late gadolinium enhancement positive/mapping positive, late gadolinium enhancement negative/mapping positive, and normal CMR. Expanding the bibliography and discussion on this topic would be useful.
Response:
- Thank you for your question. We have data on 68 cardiac magnetic resonance imaging (CMR) scans that confirmed the diagnosis of MINOCA in our cohort. However, further detailed phenotypic classification, such as late gadolinium enhancement positivity or mapping, was not specifically collected. We have data from a total of 68 patients with CMR. This is ultimately a limitation of the study, that no systematic implementation of this investigation was carried out. LGE was positive in 32 patients and negative in 36 patients.
- How were the age cut-offs determined for the population division? I suggest also considering a division based on the median age of the population.
Response:
- We divided the patients in four groups in quartiles as required, see new statistic below. The first division is based on the number of investigated patients to have comparable number of patients in all groups. However, the results from your suggestion are comparable as compared to the old division of age groups.
Table 1: Baseline characteristics of 373 patients initially presenting Troponin-positive with non-obstructive coronary artery disease according to age |
||||||
Variables |
All patients n=373 |
<52 n=88 |
52-64 n=95 |
65-75 n=91 |
>75 n=99 |
P Value |
Age – years, mean+SD |
63+15.6 |
41±9.4 |
59±3.5 |
70±2.8 |
80±3.8
|
<0.001 |
Male – no, (%) |
185 (49.6) |
63 (71.6) |
52 (54.7) |
33 (36.3) |
37 (37.4) |
<0.001 |
BMI – kg/m2, mean+SD |
27.6+5.6 |
27.9±5.9 |
27.8±6 |
27.1±5.2 |
27.7±5.3 |
0.781 |
Duration of hospitalization – days, mean+SD |
10+8.5 |
8±9.3 |
10.3±10.1 |
10.9±9.2 |
10.8±8.7
|
0.068 |
Symptoms – no, (%) |
|
|
|
|
|
|
Angina pectoris |
226 (61.6) |
63 (71.6) |
56 (59) |
50 (55) |
57 (57.6) |
0.099 |
Dyspnea |
164 (44.4) |
21 (23.9) |
49 (51.6) |
46 (50.6) |
48 (48.5) |
<0.001 |
Palpations |
45 (12.3) |
13 (14.8) |
8 (8.4) |
15 (16.5) |
9 (9.1) |
0.246 |
Clinic parameter, mean + SD |
|
|
|
|
|
|
Systolic BP, mmHg |
146.6+64.4 |
138±25.5 |
141±31.2 |
161.8±11.7 |
144.6±27.6 |
0.126 |
Diastolic BP, mmHg |
84.8+18.1 |
86.8±17.1 |
84±22.4 |
85±15.2 |
80.7±16.6 |
0.137 |
Heart rate, bpm |
89.8+28.6 |
84.5±23.5 |
92±29.7 |
92.8±31.3 |
86.3±29.2 |
0.298 |
ECG – no, (%) |
|
|
|
|
|
|
ST elevation |
55 (14.8) |
28 (31.8) |
7 (7.4) |
7 (7.7) |
13 (13.1) |
<0.001 |
Inversed T-Waves |
183 (49.2) |
28 (31.8) |
49 (51.6) |
50 (55) |
56 (56.6) |
0.002 |
Medical history – no, (%) |
|
|
|
|
|
|
Current Smoking |
85 (23.1) |
34 (39.1) |
27 (28.4) |
16 (17.6) |
8 (8.1) |
<0.001 |
Obesity – no, (%) |
112 (30) |
27 (30.7) |
30 (31.6) |
25 (27.5) |
30 (30.3) |
0.938 |
Arterial hypertension |
253 (68.2) |
37 (42.1) |
60 (63.2) |
73 (80.2) |
83 (83.8) |
<0.001 |
Dyslipidemia |
99 (26.6) |
18 (20.5) |
25 (26.3) |
24 (26.4) |
32 (32.3) |
0.341 |
Diabetes Mellitus |
65 (17.5) |
8 (9.1) |
16 (16.8) |
15 (16.5) |
26 (26.3) |
0.021 |
COPD |
47 (12.6) |
1 (1.1) |
13 (13.7) |
19 (20.9) |
14 (14.1) |
<0.001 |
Bronchial Asthma |
33 (8.9) |
8 (9.1) |
11 (11.6) |
6 (6.6) |
8 (8.1) |
0.663 |
Malignancy |
47 (12.7) |
2 (2.3) |
8 (8.4) |
18 (19.8) |
19 (19.2) |
<0.001 |
Kidney disease |
53 (14.3) |
5 (5.7) |
12 (12.6) |
8 (8.8) |
28 (28.3) |
<0.001 |
Neurological disease |
90 (24.3) |
12 (13.6) |
20 (21.1) |
23 (25.3) |
35 (35.4) |
0.006 |
Autoimmune disease |
17 (4.6) |
5 (5.7) |
5 (5.3) |
0 (0) |
7 (7.1) |
0.103 |
Psychiatric disease |
39 (10.5) |
8 (9.1) |
16 (16.8) |
8 (8.8) |
7 (7.1) |
0.114 |
Pacemaker |
14 (3.8) |
0 (0) |
2 (2.1) |
5 (5.5) |
7 (7.1) |
0.047 |
History of Malignant cardiac arrhythmias* |
8 (2.2) |
0 (0) |
0 (0) |
4 (4.4) |
4 (4) |
0.049 |
Bradycardic arrhythmias |
6 (1.6) |
0 (0) |
0 (0) |
3 (3.3) |
3 (3) |
0.119 |
- AV block 2 Mobitz |
3 (0.8) |
0 (0) |
0 (0) |
2 (2.2) |
1 (1) |
0.291 |
- AV block 3 |
3 (0.8) |
0 (0) |
0 (0) |
1 (1.1) |
2 (2) |
0.339 |
- Asystole |
0 (0) |
0 (0) |
0 (0) |
0 (0) |
0 (0) |
0 |
Ventricular arrhythmias |
1 (0.3) |
0 (0) |
0 (0) |
1 (1.1) |
0 (0) |
0.381 |
- sustained |
1 (0.3) |
0 (0) |
0 (0) |
1 (1.1) |
0 (0) |
0.381 |
- non-sustained |
0 (0) |
0 (0) |
0 (0) |
0 (0) |
0 (0) |
0 |
- ventricular fibrillation |
0 (0) |
0 (0) |
0 (0) |
0 (0) |
0 (0) |
0 |
Torsade de pointes |
1 (0.3) |
0 (0) |
0 (0) |
0 (0) |
1 (1) |
0.434 |
Atrial fibrillation |
57 (15.4) |
1 (1.4) |
9 (9.5) |
12 (13.3) |
16 (16.2) |
0.005 |
Laboratory values, mean + SD |
|
|
|
|
|
|
Troponin (µg/l) |
2.1+8.4 |
2.7±5.2 |
0.1±14.5 |
1.6±6.7 |
0.2±0.6 |
0.129 |
Creatin Phosphokinase (U/l) |
288.6+548.9 |
5.7±6.5 |
6.6±14.3 |
3.8±5.6 |
463.5 ±4553.2 |
0.446 |
BNP (pg/ml) |
65.1±123 |
27.7±61.5 |
61.2±101.6 |
48.8±199 |
37.5±29 |
0.003 |
Creatinine (mg/dl) |
93.5±45.7 |
82.6±17.9 |
79.2±71.9 |
90±32 |
103.4±40.4 |
0.017 |
TSH (mU/l) |
2+1.7 |
2±1.2 |
1.5±2.5 |
1.5±1.2 |
1.9±1.3 |
0.114 |
fT3 (pmol/l) |
5.2+1 |
5.5±1 |
5.3±1.1 |
4.9±1.1 |
4.7±0.6 |
0.087 |
fT4 (ng/l) |
10.6±2.8 |
13.4±2.3 |
11.7±2.2 |
14.1±4.6 |
14.5±3.5 |
0.029 |
Echocardiography data, n (%) |
|
|
|
|
|
|
LVEF %, (on admission), mean + SD |
36.9±25.7 |
56±21.8
|
33.2±26.3 |
34.9±26.3
|
32.7±25.4 |
<0.001 |
- LVEF > 50 % |
170 (66.9) |
58 (65.9) |
36 (37.9) |
37 (41.1) |
39 (39.4) |
0.035 |
- LVEF 40-49 % |
31 (12.2) |
5 (5.7) |
9 (9.5) |
6 (6.7) |
11 (11.1) |
0.211 |
- LVEF < 40 % |
52 (20.5) |
9 (10.2) |
12 (12.6) |
18 (20) |
13 (13.1) |
0.133 |
Left ventricular hypertrophy |
103 (29.1) |
17 (19.5) |
27 (28.4) |
24 (26.7) |
35 (35.4) |
0.044 |
Tricuspid valve regurgitation |
89 (24.2) |
8 (9.1) |
25 (27.4) |
22 (24.4) |
34 (34.3) |
<0.001 |
- mild |
63 (17.1) |
6 (6.8) |
20 (21.1) |
13 (14.4) |
24 (24.2) |
0.007 |
- moderate |
23 (6.3) |
2 (2.3) |
4 (4.2) |
9 (10) |
8 (8.1) |
0.127 |
- severe |
3 (0.8) |
0 (0) |
1 (1.1) |
0 (0) |
2 (2) |
0.341 |
Mitral valve regurgitation |
106 (28.8) |
9 (10.2) |
22 (23.2) |
31 (34.4) |
44 (44.4) |
<0.001 |
- mild |
77 (20.9) |
8 (9.1) |
17 (17.9) |
20 (22.2) |
32 (32.3) |
0.001 |
- moderate |
19 (5.2) |
1 (1.1) |
3 (3.2) |
8 (8.8) |
7 (7.1) |
0.077 |
- severe |
10 (2.7) |
0 (0) |
2 (2.1) |
3 (3.3) |
5 (5) |
0.188 |
Aortic valve regurgitation |
39 (10.6) |
2 (2.3) |
7 (7.4) |
11 (12.1) |
19 (19.2) |
0.001 |
- mild |
31 (8.4) |
2 (2.3) |
5 (5.3) |
9 (9.9) |
15 (15.2) |
0.008 |
- moderate |
4 (1.1) |
0 (0) |
2 (2.1) |
1 (1.1) |
1 (1) |
0.579 |
- severe |
4 (1.1) |
0 (0) |
0 (0) |
1 (1.1) |
3 (3) |
0.137 |
Drugs on admission, n (%) |
|
|
|
|
|
|
ß-Blocker |
131 (35.2) |
11 (12.5) |
33 (34.7) |
38 (41.8) |
49 (49.5) |
<0.001 |
ACE inhibitor |
121 (32.6) |
16 (18.2) |
30 (31.6) |
32 (35.2) |
43 (43.4) |
0.002 |
Sartans |
57 (15.3) |
4 (4.6) |
15 (15.8) |
15 (16.5) |
23 (23) |
0.005 |
Ca-Blocker |
74 (19.9) |
7 (8) |
15 (15.8) |
18 (19.8) |
34 (34.3) |
<0.001 |
Diuretics |
101 (27.2) |
11 (12.5) |
19 (20) |
19 (20.9) |
52 (52.5) |
<0.001 |
Statin |
281 (76.8) |
10 (11.5) |
16 (16.8) |
24 (26.4) |
35 (35.4) |
<0.001 |
Anticoagulants** |
58 (15.6) |
1 (1.1) |
9 (9.6) |
19 (20.9) |
29 (29.3) |
<0.001 |
Aspirin |
79 (21.2) |
7 (8) |
25 (26.3) |
17 (18.7) |
30 (30.3) |
<0.001 |
Clopidogrel |
18 (4.8) |
2 (2.3) |
5 (5.3) |
4 (4.4) |
7 (7.1) |
0.494 |
Prasugrel |
0 (0) |
0 (0) |
0 (0) |
0 (0) |
0 (0) |
0 |
Antiarrhythmics*** |
10 (2.7) |
1 (1.1) |
3 (3.2) |
4 (4.4) |
2 (2) |
0.558 |
p values for the comparison between groups of ages; SD, Standard deviation; BMI, body mass index; BP, blood pressure; ECG, Electrocardiogram; COPD, Chronic obstructive pulmonary disease; AV, atrioventricular; BNP, brain natriuretic Peptide; LVEF, left ventricular ejection fraction; ACE, Angiotensin-converting-enzyme; *, only one malignant cardiac/supraventricular arrhythmia is counted per patient (even if one patient has several arrhythmias at the same time); ** Coumarin, heparin, selective factor 10-blocker, direct thrombin inhibitors; ***, Ivabradine, Flecainide, Sotalol, Dronedarone, Digitalis. |
Table 2: Medication at discharge |
||||||
|
All patients n=373 |
<52 n=88 |
52-64 n=95 |
65-75 n=91 |
>75 n=99 |
P Value |
ß-Blocker |
270 (72.4) |
57 (64.8) |
74 (77.9) |
69 (75.8) |
70 (70.7) |
0.199 |
ACE inhibitor |
224 (60.1) |
54 (61.4) |
59 (62.1) |
54 (59.3) |
57 (57.6) |
0.919 |
Sartans |
62 (16.6) |
6 (6.8) |
17 (17.9) |
18 (19.8) |
21 (21.2) |
0.039 |
Ca Blocker |
100 (26.8) |
14 (15.9) |
23 (24.2) |
24 (26.4) |
39 (39.4) |
0.003 |
Diuretics |
166 (44.5) |
16 (18.2) |
39 (41.1) |
45 (49.5) |
66 (66.7) |
<0.001 |
Statin |
206 (55.2) |
27 (30.7) |
42 (44.2) |
42 (46.2) |
56 (56.6) |
0.005 |
Anticoagulants* |
104 (27.9) |
5 (5.7) |
25 (26.3) |
34 (37.4) |
40 (40.4) |
<0.001 |
Aspirin |
175 (46.9) |
30 (34.1) |
50 (52.6) |
43 (47.3) |
52 (52.5) |
0.040 |
Clopidogrel |
68 (18.2) |
15 (17.1) |
19 (20) |
14 (15.4) |
20 (20.2) |
0.795 |
Prasugrel |
0 (0) |
0 (0) |
0 (0) |
0 (0) |
0 (0) |
0 |
Antiarrhythmics** |
26 (7) |
2 (2.3) |
6 (6.3) |
12 (13.2) |
6 (6.1) |
0.035 |
ACE, Angiotensin-converting-enzyme; * Coumarin, Heparin, selective factor 10-blocker, direct thrombin inhibitors; ** Ivabradine, Flecainide, Sotalol, Dronedarone, Digitalis. |
Table 3: In-hospital complications according to age |
||||||
|
All patients n=373 |
<52 n=88 |
52-64 n=95 |
65-75 n=91 |
>75 n=99 |
P Value |
Adverse event |
132 (35.5) |
14 (15.9) |
37 (39) |
36 (39.6) |
45 (45.5) |
<0.001 |
CPR |
7 (1.9) |
0 (0) |
4 (4.2) |
1 (1.1) |
2 (2) |
0.188 |
Left ventricular thrombus |
3 (0.8) |
1 (1.1) |
1 (1.1) |
0 (0) |
1 (1) |
0.807 |
Thromboembolic event |
3 (0.8) |
1 (1.1) |
1 (1.1) |
0 (0) |
1 (1) |
0.807 |
Pulmonary edema |
9 (2.7) |
1 (1.1) |
4 (4.2) |
2 (2.2) |
2 (2) |
0.576 |
Cardiogenic shock |
9 (2.7) |
1 (1.1) |
4 (4.2) |
2 (2.2) |
2 (2) |
0.576 |
Invasive ventilation |
29 (7.8) |
4 (4.6) |
12 (12.6) |
7 (7.7) |
6 (6) |
0.187 |
Non-invasive ventilation |
11 (3) |
1 (1.1) |
1 (1.1) |
6 (6.6) |
3 (3) |
0.093 |
Stroke |
1 (0.3) |
0 (0) |
1 (1.1) |
0 (0) |
0 (0) |
0.404 |
Malignant cardiac arrhythmias* (On admission/in hospital) |
39 (10.5) |
8 (9.1) |
14 () |
8 (8.8) |
9 (9.1) |
0.478 |
Bradycardic arrhythmias |
13 (3.5) |
3 (3.4) |
2 (2.1) |
4 (4.49 |
4 (4) |
0.838 |
- AV block 2 Mobitz |
3 (0.8) |
2 (2.3) |
0 (0) |
1 (1.1) |
0 (0) |
0.258 |
- AV block 3 |
1 (0.3) |
0 (0) |
0 (0) |
0 (0) |
1 (1) |
0.430 |
- Asystole |
10 (2.7) |
2 (2.3) |
2 (2.1) |
3 (3.3) |
3 (3) |
0.950 |
Ventricular arrhythmias |
14 (3.8) |
2 (2.3) |
6 (6.3) |
3 (3.3) |
3 (3) |
0.486 |
- sustained |
8 (2.2) |
2 (2.3) |
2 (2.1) |
2 (2.2) |
2 (2) |
0.999 |
- non-sustained |
6 (1.6) |
0 (0) |
4 (4.2) |
1 (1.1) |
1 (1) |
0.114 |
- ventricular fibrillation |
16 (4.3) |
3 (3.4) |
9 (9.5) |
1 (1.1) |
3 (3) |
0.029 |
Torsade de pointes |
2 (0.5) |
0 (0) |
1 (1.1) |
0 (0) |
1 (1) |
0.604 |
Supraventricular arrhythmias* |
85 (22.8) |
4 (4.6) |
19 (20) |
27 (29.7) |
35 (35.4) |
<0.001 |
Atrial fibrillation |
77 (20.6) |
3 (3.4) |
14 (14.7) |
26 (28.6) |
34 (34.3) |
<0.001 |
- first appearance |
43 (11.5) |
3 (3.4) |
7 (7.4) |
11 (12.1) |
22 (12.1) |
<0.001 |
- recurrent |
34 (9.1) |
0 (0) |
7 (7.4) |
15 (16.5) |
12 () |
<0.001 |
Atrial flutter |
9 (2.4) |
1 (1.1) |
5 (5.3) |
2 (2.2) |
1 (1) |
0.194 |
- first appearance |
9 (2.4) |
1 (1.1) |
5 (5.3) |
2 (2.2) |
1 (1) |
0.194 |
- recurrent |
0 (0) |
0 (0) |
0 (0) |
0 (0) |
0 (0) |
0 |
In-hospital death |
10 (2.7) |
0 (0) |
4 (4.2) |
3 (3.3) |
3 (3) |
0.329 |
Cardiac caused death |
6 (1.6) |
0 (0) |
3 (3.2) |
2 (2.2) |
1 (1) |
0.349 |
Non-cardiac caused death |
4 (1.1) |
0 (0) |
1 (1.1) |
1 (1.1) |
2 (2) |
0.619 |
Adverse event, major adverse cardiac and cerebrovascular events; CPR, cardiopulmonary resuscitation; ECMO, extracorporeal membrane oxygenation; AV, atrioventricular; *, only one malignant cardiac/supraventricular arrhythmia is counted per patient (even if one patient has several arrhythmias at the same time). |
Table 4: in-hospital treatment approach |
||||||
|
All patients n=373 |
<52 n=88 |
52-64 n=95 |
65-75 n=91 |
>75 n=99 |
P Value |
FU EF (%) |
49+14.7 |
60±12,1 |
49.2±15 |
42.2±15.1 |
43.8±12 |
<0.001 |
- LVEF >50% |
100 (26.8) |
37 (42.1) |
28 (29.5) |
13 (14.3) |
22 (22.2) |
<0.001 |
- LVEF 40-49% |
27 (7.2) |
5 (5.7) |
6 (6.3) |
6 (6.6) |
10 (10.1) |
0.636 |
- LVEF <40% |
50 (13.4) |
6 (6.8) |
11 (11.6) |
19 (20.9) |
14 (14.1) |
0.037 |
Cardioversion |
50 (13.9) |
6 (6.8) |
21 (22.1) |
20 (22) |
17 (17.2) |
0.019 |
Pacemaker |
64 (17.2) |
2 (2.3) |
11 (11.6) |
5 (5.5) |
6 (6.1) |
0.076 |
Life Vest |
24 (6.5) |
6 (6.8) |
4 (4.2) |
2 (2.2) |
0 (0) |
0.056 |
FU EF, follow up ejection fraction; LVEF, left ventricular ejection fraction. |
Table 5: Clinical Outcome according to age at follow-up. |
||||||
|
All patients n=288 |
<52 n=68 |
52-64 n=72 |
65-75 n=65 |
>75 n=83 |
P Value |
Adverse event |
100 (34.8) |
4 (5.9) |
16 (22.2) |
25 (38.5) |
53 (63.9) |
<0.001 |
Stroke |
10 (4.2) |
2 (2.9) |
2 (2.8) |
1 (1.5) |
5 (6) |
0.398 |
Thromboembolic event |
6 (2.5) |
2 (2.9) |
0 (0) |
1 (1.5) |
3 (3.6) |
0.229 |
Recurrence of Troponin-positive with non-obstructive CAD |
3 (1.3) |
0 (0) |
1 (1.4) |
2 (3.1) |
0 (0) |
0.253 |
Cardiac arrest |
4 (1.7) |
0 (0) |
1 (1.4) |
2 (3.1) |
1 (1.2) |
0.473 |
Percutaneous coronary intervention |
17 (7.2) |
2 (2.9) |
4 (5.6) |
6 (9.2) |
5 (6) |
0.266 |
Death |
69 (24.1) |
0 (0) |
10 (13.9) |
17 (26.2) |
42 (50.6) |
<0.001 |
- cardiac caused death |
5 (2.1) |
0 (0) |
0 (0) |
3 (4.6) |
2 (2.4) |
0.062 |
- non-cardiac caused death |
12 (5.1) |
0 (0) |
5 (6.9) |
1 (1.5) |
6 (7.2) |
0.012 |
Adverse event, major adverse cardiac and cerebrovascular events; CAD, coronary artery disease; NSTEMI, Non-ST-segment elevation myocardial infarction; STEMI, ST-segment elevation myocardial infarction; CPR, cardiopulmonary resuscitation |
How do the authors explain such a low left ventricular ejection fraction upon admission? Were other categories of patients included (e.g., takotsubo)?
Response:
- Thank you for your question. Among the study population, 22 patients (5.9%) were diagnosed with takotsubo syndrome (TTS), a condition known to cause transient left ventricular dysfunction and low ejection fraction upon admission. Additionally, 48 patients (12.9%) had myocarditis, which can also contribute to reduced left ventricular ejection fraction (LVEF) during the acute phase of the disease. Importantly, in several cases, the low LVEF observed at admission improved during follow-up, likely reflecting recovery from these reversible or acute conditions.
The population was largely under-treated. Could this have influenced the outcomes? I recommend expanding the discussion on this point, considering that patients with MINOCA are at high risk for reinfarction and that managing low-density lipoprotein levels is mandatory (cite: 10.1016/j.pcad.2023.10.006).
Response:
- We conducted a comprehensive investigation into the impact of demographics, clinical characteristics, ECG findings, medical history, laboratory values, and medications on both in-hospital and long-term outcomes. In the univariate analysis, some drugs were identified as a predictor for in-hospital outcomes (adverse events and in-hospital mortality), but this association was not confirmed in the multivariate analysis. Similarly, certain medications emerged as predictors for long-term outcomes in the univariate analysis but lost significance in the multivariate model (refer to Tables 6 and 7). We appreciate your suggestion and have expanded the discussion to address the potential influence of under-treatment on patient outcomes. Specifically, we cited the recommended reference (10.1016/j.pcad.2023.10.006) highlighting the high risk of reinfarction in patients with MINOCA, often due to the progression of atherosclerosis that necessitates revascularization. Furthermore, LDL-C management appeared to be suboptimal in these patients.
Table 6: univariate and multiple Logistic Regression Analysis for in-hospital outcome |
||||||||
Variable |
Univariate analysis |
|
Multivariable analysis |
|
||||
OR |
95% CI |
P value |
OR |
95% CI |
P value |
|||
Age |
1.03 |
1.018-1.050 |
<0.001 |
|
1.00 |
0.959-1.047 |
0.919 |
|
Male |
1.24 |
0.809-1.892 |
0.327 |
|
|
|
|
|
BMI |
1.01 |
0.974-1.052 |
0.537 |
|
|
|
|
|
Symptoms |
|
|
|
|
|
|
|
|
Angina pectoris |
0.24 |
0.150-0.373 |
<0.001 |
|
0.19 |
0.054-0.627 |
0.007 |
|
Dyspnea |
1.56 |
1.086-2.574 |
0.020 |
|
2.20 |
0.524-8-955 |
0.270 |
|
Palpations |
3.33 |
1.753-6.324 |
<0.001 |
|
7.68 |
1.377-42.839 |
0.020 |
|
Clinic parameter |
|
|
|
|
|
|
|
|
Systolic BP |
1.00 |
0.987-1.004 |
0.280 |
|
|
|
|
|
Diastolic BP |
0.10 |
1.012-1.026 |
0.097 |
|
|
|
|
|
Heart rate |
1.03 |
1.021-1.040 |
<0.001 |
|
1.04 |
1.016-1.072 |
0.002 |
|
ECG Data |
|
|
|
|
|
|
|
|
ST-segment elevation |
0.64 |
0.341-1.216 |
0.175 |
|
|
|
|
|
Inversed T-Waves |
0.74 |
0.481-1.131 |
0.162 |
|
|
|
|
|
Medical history |
|
|
|
|
|
|
|
|
Current Smoker |
0.67 |
0.392-1.133 |
0.134 |
|
|
|
|
|
Diabetes mellitus |
1.39 |
0.803-2.397 |
0.241 |
|
|
|
|
|
Obesity (BMI > 30 kg/m2) |
1.20 |
0.761-1.906 |
0.427 |
|
|
|
|
|
Arterial Hypertension |
2.31 |
1.407-3.806 |
<0.001 |
|
2.22 |
0.453-10.942 |
0.325 |
|
Pulmonary disease |
1.492 |
0.895-2.490 |
0.125 |
|
|
|
|
|
Dyslipidemia |
0.89 |
0.549-1.451 |
0.647 |
|
|
|
|
|
Malignancy |
1.31 |
0.702-2.458 |
0.393 |
|
|
|
|
|
Neurological disease |
1.33 |
0.813-2.163 |
0.258 |
|
|
|
|
|
Kidney disease |
1.50 |
0.830-2.704 |
0.180 |
|
|
|
|
|
Autoimmune disease |
2.17 |
0.815-5.757 |
0.121 |
|
|
|
|
|
Psychiatric disease |
0.80 |
0.391-1.635 |
0.540 |
|
|
|
|
|
Malignant cardiac arrhythmias |
1.11 |
0.261-4.721 |
0.887 |
|
|
|
|
|
Supraventricular arrhythmias |
5.86 |
3.157-10.864 |
<0.001 |
|
3.70 |
0.721-19-023 |
0.117 |
|
Laboratory values |
|
|
|
|
|
|
|
|
Troponin |
1.00 |
0.999-1.000 |
0.164 |
|
|
|
|
|
Creatine phosphokinase |
1.00 |
1.000-1.002 |
0.657 |
|
|
|
|
|
BNP |
1.00 |
1.000-1.003 |
0.037 |
|
1.00 |
0.999-1.017 |
0.095 |
|
Creatinine |
1.02 |
1.009-1.023 |
<0.001 |
|
1.03 |
1.001-1.050 |
0.042 |
|
TSH |
1.12 |
0.891-1.287 |
0.094 |
|
|
|
|
|
fT3 |
0.83 |
0.534-1.284 |
0.399 |
|
|
|
|
|
fT4 |
1.04 |
0.896-1.214 |
0.587 |
|
|
|
|
|
Echocardiography data, n (%) |
|
|
|
|
|
|
|
|
LVEF ≥50% |
0.29 |
0.169-0.505 |
<0.001 |
|
3.38 |
0.624-18.301 |
0.158 |
|
LVEF 40-49% |
2.35 |
1.099-5.020 |
0.028 |
|
6.48 |
0.862-48.796 |
0.069 |
|
LVEF ≤ 40 % |
2.72 |
1.459-5.066 |
0.002 |
|
0.42 |
0.058-2.966 |
0.381 |
|
Tricuspid valve regurgitation |
2.86 |
1.752-4.673 |
<0.001 |
|
2.35 |
0.451-12.271 |
0.310 |
|
Mitral valve regurgitation |
2.74 |
1.719-4.368 |
<0.001 |
|
0.29 |
0.046-1.762 |
0.177 |
|
Aortic valve regurgitation |
1.68 |
0.862-3.289 |
0.127 |
|
|
|
|
|
Drugs on admission |
|
|
|
|
|
|
|
|
ß-Blocker |
1.92 |
1.237-2.990 |
0.004 |
|
1.61 |
0.435-5.963 |
0.476 |
|
ACE inhibitor |
1.28 |
0.814-2.005 |
0.286 |
|
|
|
|
|
Sartane |
2.16 |
1.223-3.826 |
0.008 |
|
0.27 |
0.053-1.374 |
0.115 |
|
Ca-Blocker |
1.43 |
0.848-2.403 |
0.180 |
|
|
|
|
|
Diuretics |
2.44 |
1.523-3.894 |
<0.001 |
|
1.56 |
0.407-5.970 |
0.518 |
|
Statin |
0.92 |
0.548-1.535 |
0.742 |
|
|
|
|
|
Anticoagulants |
3.17 |
1.784-5.631 |
<0.001 |
|
2.42 |
0.416-14.051 |
0.326 |
|
Aspirin |
0.65 |
0.377-1.125 |
0.124 |
|
|
|
|
|
Antiarrhythmics |
1.23 |
0.342-4.452 |
0.749 |
|
|
|
|
|
OR, Odds ratio; CI, confidence interval; BP, blood pressure; ECG, Electrocardiogram; BMI, body mass index; BNP, brain natriuretic Peptide; LVEF, left ventricular ejection fraction; ACE, Angiotensin-converting-enzyme |
Table 7: univariate and multiple Logistic Regression Analysis for long-term outcome |
||||||||
Variable |
Univariate analysis |
|
Multivariable analysis |
|
||||
OR |
95% CI |
P value |
OR |
95% CI |
P value |
|||
Age |
1.09 |
1.060-1.112 |
<0.001 |
|
1.07 |
1.013-1.137 |
0.017 |
|
Male |
0.55 |
0.333-0.898 |
0.017 |
|
0.28 |
0.064-1.252 |
0.096 |
|
BMI |
0.99 |
0.727-1.351 |
0.954 |
|
|
|
|
|
Symptoms |
|
|
|
|
|
|
|
|
Angina pectoris |
0.85 |
0.511-1.421 |
0.540 |
|
|
|
|
|
Dyspnea |
2-39 |
1.448-3.954 |
<0.001 |
|
3.09 |
0.753-12.694 |
0.117 |
|
Palpations |
0.52 |
0.216-1.248 |
0.143 |
|
|
|
|
|
Clinic parameter |
|
|
|
|
|
|
|
|
Systolic BP |
1.00 |
0.991-1.004 |
0.505 |
|
|
|
|
|
Diastolic BP |
0.99 |
0.976-1.007 |
0.296 |
|
|
|
|
|
Heart rate |
1.00 |
0.991-1.008 |
0.922 |
|
|
|
|
|
ECG Data |
|
|
|
|
|
|
|
|
ST-segment elevation |
0.92 |
0.453-1.881 |
0.825 |
|
|
|
|
|
Inversed T-Waves |
1.86 |
1.131-3.062 |
0.015 |
|
1.33 |
0.391-4.511 |
0.649 |
|
Medical history |
|
|
|
|
|
|
|
|
Current Smoker |
0.44 |
0.230-0.845 |
0.014 |
|
1.24 |
0.241-6.383 |
0.798 |
|
Diabetes mellitus |
2.51 |
1.362-4.624 |
0.003 |
|
14.00 |
2.181-89.933 |
0.005 |
|
Obesity (BMI > 30 kg/m2) |
1.33 |
0.792-2.231 |
0.281 |
|
|
|
|
|
Arterial Hypertension |
1.78 |
1.021-3.091 |
0.042 |
|
0.50 |
0.079-3.113 |
0.454 |
|
Pulmonary disease |
2.13 |
1.187-3.836 |
0.011 |
|
0.43 |
0.077-2.378 |
0.331 |
|
Dyslipidemia |
1.18 |
0.685-2.045 |
0.546 |
|
|
|
|
|
Malignancy |
1.34 |
0.705-2.527 |
0.375 |
|
|
|
|
|
Neurological disease |
3.15 |
1.811-5.461 |
<0.001 |
|
0.80 |
0.207-3.072 |
0.743 |
|
Kidney disease |
2.21 |
1.154-4.235 |
0.017 |
|
|
|
||
Autoimmune disease |
0.88 |
0.298-2.622 |
0.825 |
|
|
|
|
|
Malignant cardiac arrhythmias |
2.64 |
0.579-12.031 |
0.210 |
|
|
|
|
|
Supraventricular arrhythmias |
2.78 |
1.387-5.553 |
0.004 |
|
|
|
|
|
Laboratory values |
|
|
|
|
|
|
|
|
Troponin |
0.99 |
0.939-1.053 |
0.848 |
|
|
|
|
|
Creatine phosphokinase |
1.00 |
1.000-1.001 |
0.724 |
|
|
|
|
|
BNP |
1.00 |
1.000-1.001 |
0.015 |
|
0.83 |
0.124-5.511 |
0.843 |
|
Creatinine |
2.00 |
1.014-3.938 |
0.045 |
|
3.64 |
0.346-38.301 |
0.282 |
|
TSH |
0.86 |
0.693-1.068 |
0.172 |
|
|
|
|
|
fT3 |
0.59 |
0.237-1.474 |
0.260 |
|
|
|
|
|
fT4 |
1.16 |
0.935-1.445 |
0.176 |
|
|
|
|
|
Echocardiography data, n (%) |
|
|
|
|
|
|
|
|
LVEF ≥50% |
0.78 |
0.418-1.439 |
0.420 |
|
|
|
|
|
LVEF 40-49% |
0.54 |
0.193-1.510 |
0.240 |
|
|
|
|
|
LVEF ≤ 40 % |
1.90 |
0.959-3.760 |
0.066 |
|
|
|
|
|
Tricuspid valve regurgitation |
0.90 |
0.512-1.583 |
0.715 |
|
|
|
|
|
Mitral valve regurgitation |
1.04 |
0.612-1.777 |
0.878 |
|
|
|
|
|
Aortic valve regurgitation |
1.03 |
0.488-2.189 |
0.931 |
|
|
|
|
|
Drugs on admission |
|
|
|
|
|
|
|
|
ß-Blocker |
1.74 |
1.054-2.880 |
0.030 |
|
0.38 |
0.092-1.558 |
0.179 |
|
ACE inhibitor |
1.78 |
1.056-2.991 |
0.030 |
|
1.78 |
0.450-7.069 |
0.410 |
|
Sartans |
1.31 |
0.684-2.511 |
0.416 |
|
|
|
|
|
Ca-Blocker |
1.61 |
0.988-3.330 |
0.055 |
|
|
|
|
|
Diuretics |
2.28 |
1.344-3.877 |
0.002 |
|
0.65 |
0.152-2.743 |
0.552 |
|
Statins |
1.03 |
0.566-1.869 |
0.925 |
|
|
|
|
|
Anticoagulants |
1.81 |
0.955-3.440 |
0.069 |
|
|
|
|
|
Aspirin |
1.89 |
1.055-3.395 |
0.032 |
|
3.33 |
0.593-18.698 |
0.172 |
|
Antiarrhythmics |
2.05 |
0.784-5.334 |
0.143 |
|
|
|
|
|
OR, Odds ratio; CI, confidence interval; BP, blood pressure; ECG, Electrocardiogram; BMI, body mass index; BNP, brain natriuretic Peptide; LVEF, left ventricular ejection fraction; ACE, Angiotensin-converting-enzyme |
Finally, considering the study's limitations and design, I recommend toning down the conclusions and interpretation of the results.
Response:
- We have carefully revised the discussion to address this concern. The conclusions and interpretations have been rephrased to reflect a more cautious and balanced tone, taking into account the study's limitations and design.
Comments on the Quality of English Language
Perform a thorough proofreading to correct grammatical issues and ensure clarity and readability.
Response:
- We have thoroughly proofread the manuscript to correct grammatical issues and enhance clarity and readability. Necessary revisions have been implemented to ensure the language meets a high standard of quality.
Reviewer 2 Report
Comments and Suggestions for Authors
The authors conducted a cohort study on 373 patients with elevated troponin but no obstructive coronary lesion, screened from 24,775 patients, grouped by age (<51, 51-60, 61-70, >70). Older patients had higher rates of major adverse cardiovascular events during hospitalization (47.2% in >70 vs. 15.4% in <51; p<0.001). Non-sustained ventricular tachycardia was most common in the 51-60 group (5.6%; p=0.027). At 11 years, cardiovascular mortality remained highest in older patients (4.2% in >70 group; p=0.042), suggesting age correlates with increased adverse outcomes in these cases. Primary limitations include the study's retrospective nature and small sample size.
Below are several suggestions to improve the manuscript.
1. Introduction: In the sentence, “However, the prognosis of this condition is not benign… compared to patients with CAD at one-year follow-up,” I suggest adding that patients with MINOCA have an incidence of heart failure similar to those with obstructive CAD but typically have preserved ejection fraction, as recently demonstrated (cite PMID: 37596114).
2. In the “study population” section, please provide a precise definition of coronary artery disease (CAD) as used in the study. If the definition is CAD > 50%, I recommend specifying "obstructive CAD" consistently throughout the manuscript.
3. Since the authors excluded only non-cardiac causes of myocardial injury, it is essential to also report the cardiac etiologies of the included patients, such as myocarditis, Takotsubo syndrome, true MINOCA, and cardiomyopathies. If patients with non-ischemic myocardial injury were excluded, this should be explicitly stated in the methods, along with the diagnostic modalities used to determine the final diagnosis (e.g., CMR).
4. Figure 1: It appears that 30 patients did not consent to participate in the study, yet the results section includes data from a total of 373 patients, seemingly counting these 30 individuals. If these patients did not consent to data usage, they should not be included in any analysis. Please clarify this aspect.
5. The primary endpoint should be clearly stated in the abstract as well.
6. Table 1: If supraventricular arrhythmias only refer to atrial fibrillation episodes, it would be clearer to report atrial fibrillation directly as the event.
7. Table 2: Please include data on statin therapy.
8. Discussion: The sentence “the occurrence of troponin elevation without an obstructive culprit lesion is more common in patients >70 years old” may be misleading. This study did not assess the incidence of the event across age groups within the general population. We do not have information on how many patients >70 years experienced troponin elevation without obstructive CAD.
9. Discussion: In the sentence “On the other side, in a systemic review, patients suffering from troponin elevation without an obstructive culprit lesion tended to be younger and female,” I would also specify that women with MINOCA aged ≤70 have a worse prognosis (higher MACE) compared to men and women with MIOCA, likely due to distinct pathophysiology affecting long-term prognosis (PMID: 37261384).
10. Please specify what cMRT stands for in the text.
Author Response
The authors conducted a cohort study on 373 patients with elevated troponin but no obstructive coronary lesion, screened from 24,775 patients, grouped by age (<51, 51-60, 61-70, >70). Older patients had higher rates of major adverse cardiovascular events during hospitalization (47.2% in >70 vs. 15.4% in <51; p<0.001). Non-sustained ventricular tachycardia was most common in the 51-60 group (5.6%; p=0.027). At 11 years, cardiovascular mortality remained highest in older patients (4.2% in >70 group; p=0.042), suggesting age correlates with increased adverse outcomes in these cases. Primary limitations include the study's retrospective nature and small sample size.
Below are several suggestions to improve the manuscript.
- Introduction: In the sentence, “However, the prognosis of this condition is not benign… compared to patients with CAD at one-year follow-up,” I suggest adding that patients with MINOCA have an incidence of heart failure similar to those with obstructive CAD but typically have preserved ejection fraction, as recently demonstrated (cite PMID: 37596114).
Answer:
- Thank you for your suggestion. We have incorporated this important finding into the introduction as follows: However, the prognosis of this condition is not benign. Patients with troponin elevation without obstructive coronary artery disease (CAD), as observed in the ACUITY trial, exhibited an increased risk of all cause-mortality compared to those with CAD at one-year follow-up [6]. Furthermore, these patients experienced a high incidence of heart failure (HF), comparable to individuals with obstructive CAD. Notably, HF in this population was predominantly characterized by preserved ejection fraction (HFpEF).
- In the “study population” section, please provide a precise definition of coronary artery disease (CAD) as used in the study. If the definition is CAD > 50%, I recommend specifying "obstructive CAD" consistently throughout the manuscript.
Answer:
- Thank you for your feedback. We have clarified the definition of coronary artery disease (CAD) in the “Study Population” section, specifying that obstructive CAD was defined as coronary artery stenosis >50%. Additionally, we have ensured consistent use of the term "obstructive CAD" throughout the manuscript to align with this definition and avoid ambiguity.
- Since the authors excluded only non-cardiac causes of myocardial injury, it is essential to also report the cardiac etiologies of the included patients, such as myocarditis, Takotsubo syndrome, true MINOCA, and cardiomyopathies. If patients with non-ischemic myocardial injury were excluded, this should be explicitly stated in the methods, along with the diagnostic modalities used to determine the final diagnosis (e.g., CMR).
Answer:
- Thank you for your question. Troponin was determined in cases of suspected acute coronary syndrome (ACS). A total of 373 patients fulfilled the inclusion criteria. To meet these criteria, patients first had to meet the modified criteria for acute myocardial infarction (AMI) according to the "Fourth Universal Definition of Myocardial Infarction". First, there had to be an increase or decrease in cardiac troponin levels, with at least one value above the 99th percentile in the laboratory. Secondly, there had to be clinical evidence of MI, as evidenced by at least one of the following conditions Symptoms of myocardial ischemia, new ischemic changes on the electrocardiogram, pathological Q waves, evidence of new loss of viable myocardium or new regional wall motion abnormalities suggesting an ischemic cause, or evidence of coronary thrombus by angiography or autopsy. As a secondary criterion, coronary artery obstruction had to be excluded in patients undergoing angiography. This was the case if the stenosis in any of the major epicardial vessels was <50% and no subsequent intervention was performed. Among the study population, 22 patients (5.9%) were diagnosed with Takotsubo syndrome (TTS), 48 patients (12.9%) had myocarditis, 219 patients (58.7%) presented with troponin elevation without obstructive CAD (suspected true MINOCA), and 12 patients (3.2%) had concomitant valve disease. We stated that in the methods.
- Figure 1: It appears that 30 patients did not consent to participate in the study, yet the results section includes data from a total of 373 patients, seemingly counting these 30 individuals. If these patients did not consent to data usage, they should not be included in any analysis. Please clarify this aspect.
Answer:
- Thank you for your observation. The 30 patients in question did not consent to participate in the follow-up portion of the study but explicitly agreed to the use of their data collected during hospitalization and from source records for research purposes. As such, their data were included in the baseline characteristics and outcomes analysis where applicable, but they were excluded from the follow-up-specific analyses. This approach aligns with ethical standards and the consent provided by the patients.
- The primary endpoint should be clearly stated in the abstract as well.
Answer:
- We stated primary endpoints (cardiovascular and non-cardiovascular mortality at long-term) in the abstract.
- Table 1: If supraventricular arrhythmias only refer to atrial fibrillation episodes, it would be clearer to report atrial fibrillation directly as the event.
Answer:
- We deleted the row for supraventricular arrhythmias and directly reported the rate of AF.
- Table 2: Please include data on statin therapy.
Answer:
- We have included data on statin therapy in Table 2. The rate of statin therapy was higher in older patients compared to younger patients.
- Discussion: The sentence “the occurrence of troponin elevation without an obstructive culprit lesion is more common in patients >70 years old” may be misleading. This study did not assess the incidence of the event across age groups within the general population. We do not have information on how many patients >70 years experienced troponin elevation without obstructive CAD.
Answer:
- Thank you for your feedback. We have revised the sentence to clarify that, within our study cohort, the occurrence of troponin elevation without an obstructive culprit lesion was observed more frequently in patients >70 years old compared to other age groups. However, we acknowledge that our study does not provide data on the incidence of this condition in the general population.
- Discussion: In the sentence “On the other side, in a systemic review, patients suffering from troponin elevation without an obstructive culprit lesion tended to be younger and female,” I would also specify that women with MINOCA aged ≤70 have a worse prognosis (higher MACE) compared to men and women with MIOCA, likely due to distinct pathophysiology affecting long-term prognosis (PMID: 37261384).
Answer:
- Thank you for your suggestion. We have incorporated the reference into our discussion: Another study highlighted that women aged ≤70 years with MINOCA had worse prognoses, with significantly higher rates of major adverse cardiovascular events (MACE) compared to men and women with MI and obstructive CAD. This difference was attributed to distinct pathophysiological mechanisms affecting long-term outcomes.
- Please specify what cMRT stands for in the text.
- We apologize for the oversight. "CMR" stands for cardiac magnetic resonance imaging. We have corrected this term throughout the manuscript.
Reviewer 3 Report
Comments and Suggestions for Authors
Authors of this original study analyzed in-hospital complications and the occurrence of adverse events in patients with elevated troponin level and non-obstructive CAD. All parts of this manuscript are technically good: abstract is good; Introduction section gives all necessary information about planned research. Methodology is generally well explained and statistical methods are adequate. Results are generally good presented; discussion is well conducted. References are adequate.
My comments are:
-
methodology inclusion/exclusion criteria: authors excluded some conditions that can cause troponin elevation, but they did not mention stress induced cardiomyopathy, myocarditis… Did authors perform some other diagnostic procedures other than coronary angiography to be sure that diagnosis in all patients is MINOCA?
-
Results: I will only suggest modifying Kaplan Meier curves (for better curve visibility) (Figure 3).
-
What about other predictors for in-hospital and long-term adverse events? As I can see, authors performed only basic analysis, not univariate and multivariate analysis that will include troponin level with other baseline, clinical, echocardiographic variables…..
Thank you
Author Response
Authors of this original study analyzed in-hospital complications and the occurrence of adverse events in patients with elevated troponin level and non-obstructive CAD. All parts of this manuscript are technically good: abstract is good; Introduction section gives all necessary information about planned research. Methodology is generally well explained and statistical methods are adequate. Results are generally good presented; discussion is well conducted. References are adequate.
My comments are:
- methodology inclusion/exclusion criteria: authors excluded some conditions that can cause troponin elevation, but they did not mention stress induced cardiomyopathy, myocarditis… Did authors perform some other diagnostic procedures other than coronary angiography to be sure that diagnosis in all patients is MINOCA?
Answer:
- Patients were required to meet the modified criteria for acute myocardial infarction (AMI) based on the "Fourth Universal Definition of Myocardial Infarction." This included: 1) A rise or fall in cardiac troponin levels with at least one value above the 99th percentile; 2)Clinical evidence of MI, such as symptoms of myocardial ischemia, new ischemic changes on ECG, pathological Q waves, new regional wall motion abnormalities, evidence of new loss of viable myocardium, or a coronary thrombus detected via angiography. Coronary artery obstruction was excluded in patients undergoing angiography if stenosis in major epicardial vessels was <50%, and no subsequent interventions were performed. Among the study population, 22 patients (5.9%) were diagnosed with Takotsubo syndrome (TTS), and 48 patients (12.9%) were diagnosed with myocarditis. The definition of MINOCA in our study followed the 2023 ACS Guidelines. Other diagnostic procedures, such as cardiac magnetic resonance (CMR), were performed in 68 patients (17.5%). However, the lack of systematic implementation of CMR remains a limitation of our study, as noted in the limitations section.
- Results: I will only suggest modifying Kaplan Meier curves (for better curve visibility) (Figure 3).
Answer:
- Thank you for your suggestion to improve the Kaplan-Meier curves for better visibility. We have made the necessary modifications, as shown in the updated Figure 3.
- What about other predictors for in-hospital and long-term adverse events? As I can see, authors performed only basic analysis, not univariate and multivariate analysis that will include troponin level with other baseline, clinical, echocardiographic variables…..
Answer:
- We conducted a comprehensive investigation into the impact of demographics, clinical characteristics, ECG findings, medical history, laboratory values, and medications on both in-hospital and long-term outcomes, please see tables below.
Table 6: univariate and multiple Logistic Regression Analysis for in-hospital outcome |
||||||||
Variable |
Univariate analysis |
|
Multivariable analysis |
|
||||
OR |
95% CI |
P value |
OR |
95% CI |
P value |
|||
Age |
1.03 |
1.018-1.050 |
<0.001 |
|
1.00 |
0.959-1.047 |
0.919 |
|
Male |
1.24 |
0.809-1.892 |
0.327 |
|
|
|
|
|
BMI |
1.01 |
0.974-1.052 |
0.537 |
|
|
|
|
|
Symptoms |
|
|
|
|
|
|
|
|
Angina pectoris |
0.24 |
0.150-0.373 |
<0.001 |
|
0.19 |
0.054-0.627 |
0.007 |
|
Dyspnea |
1.56 |
1.086-2.574 |
0.020 |
|
2.20 |
0.524-8-955 |
0.270 |
|
Palpations |
3.33 |
1.753-6.324 |
<0.001 |
|
7.68 |
1.377-42.839 |
0.020 |
|
Clinic parameter |
|
|
|
|
|
|
|
|
Systolic BP |
1.00 |
0.987-1.004 |
0.280 |
|
|
|
|
|
Diastolic BP |
0.10 |
1.012-1.026 |
0.097 |
|
|
|
|
|
Heart rate |
1.03 |
1.021-1.040 |
<0.001 |
|
1.04 |
1.016-1.072 |
0.002 |
|
ECG Data |
|
|
|
|
|
|
|
|
ST-segment elevation |
0.64 |
0.341-1.216 |
0.175 |
|
|
|
|
|
Inversed T-Waves |
0.74 |
0.481-1.131 |
0.162 |
|
|
|
|
|
Medical history |
|
|
|
|
|
|
|
|
Current Smoker |
0.67 |
0.392-1.133 |
0.134 |
|
|
|
|
|
Diabetes mellitus |
1.39 |
0.803-2.397 |
0.241 |
|
|
|
|
|
Obesity (BMI > 30 kg/m2) |
1.20 |
0.761-1.906 |
0.427 |
|
|
|
|
|
Arterial Hypertension |
2.31 |
1.407-3.806 |
<0.001 |
|
2.22 |
0.453-10.942 |
0.325 |
|
Pulmonary disease |
1.492 |
0.895-2.490 |
0.125 |
|
|
|
|
|
Dyslipidemia |
0.89 |
0.549-1.451 |
0.647 |
|
|
|
|
|
Malignancy |
1.31 |
0.702-2.458 |
0.393 |
|
|
|
|
|
Neurological disease |
1.33 |
0.813-2.163 |
0.258 |
|
|
|
|
|
Kidney disease |
1.50 |
0.830-2.704 |
0.180 |
|
|
|
|
|
Autoimmune disease |
2.17 |
0.815-5.757 |
0.121 |
|
|
|
|
|
Psychiatric disease |
0.80 |
0.391-1.635 |
0.540 |
|
|
|
|
|
Malignant cardiac arrhythmias |
1.11 |
0.261-4.721 |
0.887 |
|
|
|
|
|
Supraventricular arrhythmias |
5.86 |
3.157-10.864 |
<0.001 |
|
3.70 |
0.721-19-023 |
0.117 |
|
Laboratory values |
|
|
|
|
|
|
|
|
Troponin |
1.00 |
0.999-1.000 |
0.164 |
|
|
|
|
|
Creatine phosphokinase |
1.00 |
1.000-1.002 |
0.657 |
|
|
|
|
|
BNP |
1.00 |
1.000-1.003 |
0.037 |
|
1.00 |
0.999-1.017 |
0.095 |
|
Creatinine |
1.02 |
1.009-1.023 |
<0.001 |
|
1.03 |
1.001-1.050 |
0.042 |
|
TSH |
1.12 |
0.891-1.287 |
0.094 |
|
|
|
|
|
fT3 |
0.83 |
0.534-1.284 |
0.399 |
|
|
|
|
|
fT4 |
1.04 |
0.896-1.214 |
0.587 |
|
|
|
|
|
Echocardiography data, n (%) |
|
|
|
|
|
|
|
|
LVEF ≥50% |
0.29 |
0.169-0.505 |
<0.001 |
|
3.38 |
0.624-18.301 |
0.158 |
|
LVEF 40-49% |
2.35 |
1.099-5.020 |
0.028 |
|
6.48 |
0.862-48.796 |
0.069 |
|
LVEF ≤ 40 % |
2.72 |
1.459-5.066 |
0.002 |
|
0.42 |
0.058-2.966 |
0.381 |
|
Tricuspid valve regurgitation |
2.86 |
1.752-4.673 |
<0.001 |
|
2.35 |
0.451-12.271 |
0.310 |
|
Mitral valve regurgitation |
2.74 |
1.719-4.368 |
<0.001 |
|
0.29 |
0.046-1.762 |
0.177 |
|
Aortic valve regurgitation |
1.68 |
0.862-3.289 |
0.127 |
|
|
|
|
|
Drugs on admission |
|
|
|
|
|
|
|
|
ß-Blocker |
1.92 |
1.237-2.990 |
0.004 |
|
1.61 |
0.435-5.963 |
0.476 |
|
ACE inhibitor |
1.28 |
0.814-2.005 |
0.286 |
|
|
|
|
|
Sartane |
2.16 |
1.223-3.826 |
0.008 |
|
0.27 |
0.053-1.374 |
0.115 |
|
Ca-Blocker |
1.43 |
0.848-2.403 |
0.180 |
|
|
|
|
|
Diuretics |
2.44 |
1.523-3.894 |
<0.001 |
|
1.56 |
0.407-5.970 |
0.518 |
|
Statin |
0.92 |
0.548-1.535 |
0.742 |
|
|
|
|
|
Anticoagulants |
3.17 |
1.784-5.631 |
<0.001 |
|
2.42 |
0.416-14.051 |
0.326 |
|
Aspirin |
0.65 |
0.377-1.125 |
0.124 |
|
|
|
|
|
Antiarrhythmics |
1.23 |
0.342-4.452 |
0.749 |
|
|
|
|
|
OR, Odds ratio; CI, confidence interval; BP, blood pressure; ECG, Electrocardiogram; BMI, body mass index; BNP, brain natriuretic Peptide; LVEF, left ventricular ejection fraction; ACE, Angiotensin-converting-enzyme |
Table 7: univariate and multiple Logistic Regression Analysis for long-term outcome |
||||||||
Variable |
Univariate analysis |
|
Multivariable analysis |
|
||||
OR |
95% CI |
P value |
OR |
95% CI |
P value |
|||
Age |
1.09 |
1.060-1.112 |
<0.001 |
|
1.07 |
1.013-1.137 |
0.017 |
|
Male |
0.55 |
0.333-0.898 |
0.017 |
|
0.28 |
0.064-1.252 |
0.096 |
|
BMI |
0.99 |
0.727-1.351 |
0.954 |
|
|
|
|
|
Symptoms |
|
|
|
|
|
|
|
|
Angina pectoris |
0.85 |
0.511-1.421 |
0.540 |
|
|
|
|
|
Dyspnea |
2-39 |
1.448-3.954 |
<0.001 |
|
3.09 |
0.753-12.694 |
0.117 |
|
Palpations |
0.52 |
0.216-1.248 |
0.143 |
|
|
|
|
|
Clinic parameter |
|
|
|
|
|
|
|
|
Systolic BP |
1.00 |
0.991-1.004 |
0.505 |
|
|
|
|
|
Diastolic BP |
0.99 |
0.976-1.007 |
0.296 |
|
|
|
|
|
Heart rate |
1.00 |
0.991-1.008 |
0.922 |
|
|
|
|
|
ECG Data |
|
|
|
|
|
|
|
|
ST-segment elevation |
0.92 |
0.453-1.881 |
0.825 |
|
|
|
|
|
Inversed T-Waves |
1.86 |
1.131-3.062 |
0.015 |
|
1.33 |
0.391-4.511 |
0.649 |
|
Medical history |
|
|
|
|
|
|
|
|
Current Smoker |
0.44 |
0.230-0.845 |
0.014 |
|
1.24 |
0.241-6.383 |
0.798 |
|
Diabetes mellitus |
2.51 |
1.362-4.624 |
0.003 |
|
14.00 |
2.181-89.933 |
0.005 |
|
Obesity (BMI > 30 kg/m2) |
1.33 |
0.792-2.231 |
0.281 |
|
|
|
|
|
Arterial Hypertension |
1.78 |
1.021-3.091 |
0.042 |
|
0.50 |
0.079-3.113 |
0.454 |
|
Pulmonary disease |
2.13 |
1.187-3.836 |
0.011 |
|
0.43 |
0.077-2.378 |
0.331 |
|
Dyslipidemia |
1.18 |
0.685-2.045 |
0.546 |
|
|
|
|
|
Malignancy |
1.34 |
0.705-2.527 |
0.375 |
|
|
|
|
|
Neurological disease |
3.15 |
1.811-5.461 |
<0.001 |
|
0.80 |
0.207-3.072 |
0.743 |
|
Kidney disease |
2.21 |
1.154-4.235 |
0.017 |
|
|
|
||
Autoimmune disease |
0.88 |
0.298-2.622 |
0.825 |
|
|
|
|
|
Malignant cardiac arrhythmias |
2.64 |
0.579-12.031 |
0.210 |
|
|
|
|
|
Supraventricular arrhythmias |
2.78 |
1.387-5.553 |
0.004 |
|
|
|
|
|
Laboratory values |
|
|
|
|
|
|
|
|
Troponin |
0.99 |
0.939-1.053 |
0.848 |
|
|
|
|
|
Creatine phosphokinase |
1.00 |
1.000-1.001 |
0.724 |
|
|
|
|
|
BNP |
1.00 |
1.000-1.001 |
0.015 |
|
0.83 |
0.124-5.511 |
0.843 |
|
Creatinine |
2.00 |
1.014-3.938 |
0.045 |
|
3.64 |
0.346-38.301 |
0.282 |
|
TSH |
0.86 |
0.693-1.068 |
0.172 |
|
|
|
|
|
fT3 |
0.59 |
0.237-1.474 |
0.260 |
|
|
|
|
|
fT4 |
1.16 |
0.935-1.445 |
0.176 |
|
|
|
|
|
Echocardiography data, n (%) |
|
|
|
|
|
|
|
|
LVEF ≥50% |
0.78 |
0.418-1.439 |
0.420 |
|
|
|
|
|
LVEF 40-49% |
0.54 |
0.193-1.510 |
0.240 |
|
|
|
|
|
LVEF ≤ 40 % |
1.90 |
0.959-3.760 |
0.066 |
|
|
|
|
|
Tricuspid valve regurgitation |
0.90 |
0.512-1.583 |
0.715 |
|
|
|
|
|
Mitral valve regurgitation |
1.04 |
0.612-1.777 |
0.878 |
|
|
|
|
|
Aortic valve regurgitation |
1.03 |
0.488-2.189 |
0.931 |
|
|
|
|
|
Drugs on admission |
|
|
|
|
|
|
|
|
ß-Blocker |
1.74 |
1.054-2.880 |
0.030 |
|
0.38 |
0.092-1.558 |
0.179 |
|
ACE inhibitor |
1.78 |
1.056-2.991 |
0.030 |
|
1.78 |
0.450-7.069 |
0.410 |
|
Sartans |
1.31 |
0.684-2.511 |
0.416 |
|
|
|
|
|
Ca-Blocker |
1.61 |
0.988-3.330 |
0.055 |
|
|
|
|
|
Diuretics |
2.28 |
1.344-3.877 |
0.002 |
|
0.65 |
0.152-2.743 |
0.552 |
|
Statins |
1.03 |
0.566-1.869 |
0.925 |
|
|
|
|
|
Anticoagulants |
1.81 |
0.955-3.440 |
0.069 |
|
|
|
|
|
Aspirin |
1.89 |
1.055-3.395 |
0.032 |
|
3.33 |
0.593-18.698 |
0.172 |
|
Antiarrhythmics |
2.05 |
0.784-5.334 |
0.143 |
|
|
|
|
|
OR, Odds ratio; CI, confidence interval; BP, blood pressure; ECG, Electrocardiogram; BMI, body mass index; BNP, brain natriuretic Peptide; LVEF, left ventricular ejection fraction; ACE, Angiotensin-converting-enzyme |
Reviewer 4 Report
Comments and Suggestions for Authors
The study examines long-term outcomes in patients with troponin elevation without obstructive coronary lesions, focusing on age-related differences over an 11-year period. Divided into four age groups, the study analyzes baseline characteristics, in-hospital complications, and cardiovascular outcomes. Results indicate that older patients experience more in-hospital adverse events and have higher long-term cardiovascular mortality compared to younger patients. However, non-sustained ventricular arrhythmias were more common in the 51-60 age group.
1. Early definitions of terms such as MINOCA, troponin elevation, and obstructive vs. non-obstructive coronary disease would improve clarity, especially for readers less familiar with these distinctions.
2. Provide more detail on patient selection criteria and exclusion parameters, particularly why certain conditions like stroke and renal failure were excluded.
3. Expand on the significance of non-sustained ventricular tachycardia (nsVT) in the 51-60 age group. Discuss why this might occur more frequently in this age group and its potential implications.
4. Discuss the baseline characteristics (e.g., hypertension, smoking rates) more thoroughly to relate them to observed outcomes and age differences in complications.
Author Response
The study examines long-term outcomes in patients with troponin elevation without obstructive coronary lesions, focusing on age-related differences over an 11-year period. Divided into four age groups, the study analyzes baseline characteristics, in-hospital complications, and cardiovascular outcomes. Results indicate that older patients experience more in-hospital adverse events and have higher long-term cardiovascular mortality compared to younger patients. However, non-sustained ventricular arrhythmias were more common in the 51-60 age group.
- Early definitions of terms such as MINOCA, troponin elevation, and obstructive vs. non-obstructive coronary disease would improve clarity, especially for readers less familiar with these distinctions.
Response:
- Thank you for your feedback. We have revised and improved the definitions of terms such as MINOCA, troponin elevation, and obstructive vs. non-obstructive coronary artery disease (CAD) in the Materials and Methods section to enhance clarity for readers less familiar with these distinctions:
MINOCA was defined in accordance with the 2023 ACS Guidelines of the European Society of Cardiology (ESC).
To meet the inclusion criteria, patients first had to fulfill the modified criteria for acute MI according to the ’’Fourth Universal Definition of Myocardial Infarction’’[11]. Additionally, there had to be clinical evidence of MI, as indicated by at least one of the following: symptoms of MI, new ischemic changes on the electrocardiogram, pathological Q waves, evidence of new loss of viable myocardium, new regional wall motion abnormalities suggestive of an ischemic cause, or evidence of coronary thrombus identified through angiography or autopsy.
Obstructive CAD was defined as the presence of one or more coronary stenoses with a diameter reduction of ≥50%. Nonobstructive CAD was further classified based on angiographic findings as stenoses <50% or no angiographic evidence of coronary disease.
- Provide more detail on patient selection criteria and exclusion parameters, particularly why certain conditions like stroke and renal failure were excluded.
Response:
- In accordance with the 2023 ACS guidelines, troponin elevation without obstructive CAD can result from underlying coronary, non-coronary cardiac, and non-cardiac causes. Non-cardiac causes include acute respiratory distress syndrome (ARDS), allergic reactions, end-stage renal failure, inflammation, pulmonary embolism, sepsis, and stroke.
In our study, patients had to meet the inclusion criteria by fulfilling the modified criteria for acute myocardial infarction (MI) as defined by the "Fourth Universal Definition of Myocardial Infarction" [11]. This required clinical evidence of MI, as indicated by at least one of the following: symptoms of MI, new ischemic changes on the electrocardiogram, pathological Q waves, evidence of new loss of viable myocardium, new regional wall motion abnormalities suggestive of an ischemic cause, or evidence of coronary thrombus identified through angiography or autopsy.
Conditions like stroke and renal failure were excluded to ensure a more specific evaluation of troponin elevation related to MINOCA and to avoid confounding factors caused by non-cardiac or systemic illnesses. Additional details are provided in the Materials and Methods section.
- Expand on the significance of non-sustained ventricular tachycardia (nsVT) in the 51-60 age group. Discuss why this might occur more frequently in this age group and its potential implications.
Response:
- Some conditions, such as Takotsubo syndrome (TTS) and hypertrophic cardiomyopathy (HCM), occurred in this age range. In one study, the median age of HCM diagnosis was 45.8 years. Another study reported that 1,194 patients (56.9%) were between 51 and 74 years old. Myocarditis occurred also in this age range. In our study, troponin levels in this group were higher compared to older age groups. Additionally, the left ventricular ejection fraction (LVEF) on admission was reported to be 37.5 ± 25, which may also contribute to the higher occurrence of non-sustained ventricular tachycardia (nsVT) in this group. These factors are discussed in greater detail in the Discussion
- Discuss the baseline characteristics (e.g., hypertension, smoking rates) more thoroughly to relate them to observed outcomes and age differences in complications..
Response:
- We conducted a comprehensive investigation into the impact of demographics, clinical characteristics, ECG findings, medical history, laboratory values, and medications on both in-hospital and long-term outcomes. In the multivariate analyses for both in-hospital and long-term outcomes, hypertension and smoking rates were not identified as predictors.
Round 2
Reviewer 1 Report
Comments and Suggestions for Authors
Congratulations to the authors on the improvements made to the manuscript following the revision.
However, in Lines 285–290, there is a lack of citations. I recommend updating the bibliography accordingly to address this issue.
Author Response
Comments and Suggestions for Authors
Congratulations to the authors on the improvements made to the manuscript following the revision.
However, in Lines 285–290, there is a lack of citations. I recommend updating the bibliography accordingly to address this issue.
Response: Thank you for pointing this out. We have added the necessary citation in Lines 285–290 and updated the bibliography accordingly to address this issue.
Reviewer 2 Report
Comments and Suggestions for Authors
Thank you to the authors for the revisions made, which I believe have enhanced the quality of the final manuscript.
I have no further comments.
Author Response
Comments and Suggestions for Authors
Thank you to the authors for the revisions made, which I believe have enhanced the quality of the final manuscript.
I have no further comments.
Response: Thank you for your positive feedback and for taking the time to review our manuscript. We are pleased to hear that the revisions have improved the quality of the work.